# A Diffusion-based Foundation Model for Irregular Spatio-Temporal Trajectories

## Abstract

Maritime mobility provides a representative yet extremely challenging setting for spatio-temporal sequence modeling. Global AIS trajectories are inherently irregular and noisy, spanning intercontinental routes as well as fine-grained near-port maneuvers, and are constrained by coastlines, shipping lanes, and regulations. These characteristics not only complicate trajectory prediction but also generalize to other irregular spatio-temporal domains such as urban mobility and autonomous navigation. Moreover, maritime applications involve diverse downstream tasks, including forecasting, imputation, and route planning, where training separate task-specific models is computationally costly and undermines scalability. This motivates the development of a foundation-level generative framework that can unify irregular sequence modeling across tasks. We propose GeoDiffusion, a diffusion-based foundation model for maritime trajectory modeling. GeoDiffusion introduces three core components: a Spatio-Temporal Offset Encoding (STOE) to robustly capture irregular sampling and missing data, a Transformer-based denoising network to learn both global and local dynamics, and a training-free conditional inference strategy that enforces geo-temporal consistency and unifies multiple tasks within a single pretrained model. Experiments on large-scale global AIS datasets demonstrate that GeoDiffusion achieves state-of-the-art performance across trajectory prediction, imputation, and planning, while generalizing robustly to unseen distributions. These results highlight GeoDiffusion as both a practical solution for maritime mobility and a blueprint for irregular spatio-temporal foundation models.

## 1 Introduction

Spatio-temporal sequence modeling lies at the heart of many domains, from human mobility (Solatorio, 2023) and urban transportation (Wu et al., 2021) to air traffic management (Qi et al., 2024), medical time series (Song et al., 2018), and maritime logistics (Rong et al., 2022; Capobianco et al., 2021). Across these domains, data often share three common characteristics: (1) irregular and sparse sampling, due to sensor failures, missing signals, or heterogeneous logging intervals; (2) multi-scale dynamics, where coarse-grained long-range transitions coexist with fine-grained local behaviors; and (3) spatio-temporal constraints, where movement is governed by both physical continuity and environmental structure. Moreover, irregular spatio-temporal domains are typically associated with a wide range of downstream tasks, including forecasting, imputation, planning, and anomaly detection. Training and maintaining separate task-specific models for each of these objectives is not only computationally expensive but also undermines model scalability and consistency across applications. This motivates the development of a foundation model that can unify diverse tasks within a single framework, enabling efficient adaptation while reducing the cost of repeated retraining.

Among these domains, global maritime trajectories represent a particularly challenging and representative case: AIS data is inherently irregular and noisy, trajectories span intercontinental scales as well as near-port maneuvers, and vessel movement is constrained by coastlines, shipping lanes, and regulations. Thus, maritime mobility not only poses extreme challenges but also provides a testbed whose solutions generalize naturally to other irregular spatio-temporal domains such as urban mobility and autonomous navigation.

Existing sequence modeling approaches have made substantial progress in spatio-temporal learning. Transformer-based models such as Informer (Zhou et al., 2021), Autoformer (Wu et al., 2021),

TimesNet (Wu et al., 2023a), and Crossformer (Zhang & Yan, 2023) improve long-range temporal forecasting through efficient attention mechanisms and decomposition strategies. Generative approaches such as GPT-ST (Li et al., 2023) extend pretraining paradigms to structured sequences. However, these models typically assume uniformly sampled data and thus fall short when confronted with the irregularity and sparsity inherent in real-world trajectories. In the maritime domain, specialized approaches such as AISFuser (Zhang et al., 2025), AIS-Hybrid (Zhu et al., 2024), and DAISTIN (Magnussen et al., 2023) address specific tasks like encounter classification or trajectory interpolation, but they are scenario-specific and lack the generalization ability required of a foundation model. Recent diffusion-based efforts, such as PG-Diffusion (Zhang et al., 2024), demonstrate promise for trajectory recovery under physics-based constraints, but are still narrowly tailored to vessel motion. Collectively, these limitations highlight a critical gap: the absence of a foundation-level generative model that consolidates irregular sequence modeling while supporting multiple downstream tasks in an efficient and consistent manner.

Building such a model requires addressing several fundamental challenges. First, real-world sequences are often irregularly sampled with missing data, making it difficult to maintain temporal continuity or infer plausible dynamics in long gaps. Second, trajectories exhibit multi-scale spatio-temporal dynamics, where global transitions over long distances must be captured alongside local maneuvering near destinations. Third, sequence generation must preserve geo-temporal consistency, ensuring that predicted paths follow physically plausible routes constrained by both environmental structures and temporal smoothness. These challenges are not limited to maritime mobility but broadly characterize irregular spatio-temporal data across diverse application areas.

To address these issues, we propose GeoDiffusion, a diffusion-based (Ho et al., 2020; Nichol & Dhariwal, 2021) foundation model for global-scale trajectory modeling. At its core, GeoDiffusion introduces a Spatio-Temporal Offset Encoding that explicitly encodes relative temporal intervals and spatial distances, enabling the model to robustly handle irregular sampling and missing observations. On top of this representation, a Transformer-based Vaswani et al. (2017) denoising network captures both global navigational patterns and fine-grained local dynamics, thereby accommodating multi-scale behaviors within a unified framework. Finally, a training-free conditional inference strategy with trajectory constraints ensures that generated sequences remain consistent with physical continuity and navigational feasibility, allowing the same pretrained model to flexibly support tasks such as trajectory prediction, imputation, and route planning without task-specific retraining.

Our choice of diffusion as the generative backbone is motivated by its natural alignment with trajectory modeling. The reverse denoising process mirrors the unfolding of vessel paths from latent navigational intent into observable movements, providing an intuitive and principled generative prior. Unlike autoregressive methods that accumulate errors over long horizons, diffusion models offer robust uncertainty handling and stable generation. Furthermore, diffusion enables a single pretrained model to generalize across multiple downstream tasks, aligning with the vision of spatio-temporal foundation models. Our contributions are summarized in threefold:

- Foundation model perspective: We introduce GeoDiffusion, the first diffusion-based generative framework for large-scale maritime trajectories, while emphasizing its broader applicability to irregular spatio-temporal domains.

- Methodological innovations: We propose Spatio-Temporal Offset Encoding for irregular time–space gaps, integrate latent cross-attention to enhance representation learning, and design a training-free conditional inference mechanism to unify multiple downstream tasks within a single model.

- Extensive validation: Pretrained on global AIS data, GeoDiffusion achieves state-of-the-art results on trajectory prediction, imputation, and route planning across private and public datasets. Notably, it generalizes robustly to unseen distributions without fine-tuning, demonstrating its potential as a general-purpose spatio-temporal foundation model.

## 2  RELATED WORK

**Trajectory Modeling and Maritime Forecasting.** Trajectory modeling has long been an active research area across domains such as autonomous driving, pedestrian forecasting, and maritime mobility. Classical approaches often rely on physics-based models (e.g., Kalman filters, dynamic

Bayesian networks), which struggle to capture multimodal behaviors and large-scale spatiotemporal patterns. Recent works apply RNNs, temporal CNNs, and attention-based networks to predict vessel or pedestrian trajectories (Alahi et al., 2016; Cui et al., 2019). In the maritime domain, AIS-based models such as AIS-Hybrid (Zhu et al., 2024) and graph-based maritime traffic predictors AISFuser (Zhang et al., 2025) have shown promising results in route classification and future waypoint estimation.

**Diffusion Models for Sequential and Spatiotemporal Data.** Diffusion probabilistic models (DPMs) have recently emerged as powerful generative frameworks, achieving state-of-the-art results in image (Ho et al., 2020), audio (Kong et al., 2021), and language generation (Lovelace et al., 2023). While most early diffusion models were designed for continuous data domains, recent works have adapted diffusion to discrete sequences via embedding techniques and latent modeling (Rombach et al., 2022). In the time-series domain, models such as TimeGrad, CSDI (Tashiro et al., 2021), and TimesNet (Wu et al., 2023b) explore probabilistic temporal modeling using either score-based dynamics or encoder-decoder diffusion. However, these models assume uniform time steps and 1D sequence structures, making them unsuitable for irregularly sampled, geospatial trajectories. Latent diffusion models (LDMs) (Rombach et al., 2022) and (Lovelace et al., 2023) improve both sampling speed and quality by operating in compressed latent spaces. Inspired by this, we adapt LDMs to the maritime setting by introducing structured spatial tokenization, a domain-specific Trajectory Encoder, and a diffusion transformer trained in latent space for flexible trajectory generation.

**Foundation Models and Pretraining for Spatiotemporal Domains.** Foundation models trained on large-scale data have revolutionized language (Brown et al., 2020), vision (Dosovitskiy et al., 2020), and multimodal learning (Radford et al., 2021). However, pretrained foundation models for mobility or geospatial movement data remain underexplored. A few recent attempts have emerged, such as TrajGPT for synthetic trajectory generation (Hsu et al., 2024), and GeoFormer (Solatorio, 2023) for urban mobility modeling. These models are either limited to local city-scale data, or rely on deterministic generation.

Our work is the first to propose a diffusion-based foundation model for maritime trajectories, trained on real-world, irregular AIS records at a global scale. By integrating latent diffusion with physically grounded spatial encoding, we demonstrate a scalable, generative model that generalizes across routes, regions, and sampling patterns.

## 3 PROBLEM FORMULATION

Let $\mathcal{X}$ denote the space of all trajectories, where each trajectory $\mathbf{x} \in \mathcal{X}$ is represented as a sequence of discrete spatiotemporal tokens: $\mathbf{x} = \{s_1, s_2, \ldots, s_N\}$, where each token $s_i = (l_i, \tau_i)$ corresponds to the discretized spatial location $l_i$ and its associated timestamp $\tau_i$. We adopt the denoising diffusion probabilistic modeling (DDPM) framework (Ho et al., 2020; Song & Ermon, 2019; Sohl-Dickstein et al., 2015) to formulate trajectory modeling as a self-supervised trajectory reconstruction task via *diffusion* in a latent space. Specifically, we define a *forward diffusion process* that gradually adds Gaussian noise to the input trajectory and a *reverse denoising process* that learns to recover the original trajectory from noise.

In the forward process, given a clean trajectory $\mathbf{x} \in \mathcal{X}$, we first encode it into a latent space: $\mathbf{z} = f^{\mathrm{enc}}(x_0)$, where $f^{\mathrm{enc}}$ is an encoder that maps the token sequence into a latent space. We then define a Markovian forward process that gradually adds Gaussian noise to produce a sequence of noisy latent variables $\{z_t\}_{t=1}^T$. The process follows a fixed noise schedule $\{\beta_t\}_{t=1}^T$, such that:

$$q(z_t \mid z_{t-1}) = \mathcal{N}(z_t; \sqrt{1 - \beta_t} \cdot z_{t-1}, \beta_t \mathbf{I}). \tag{1}$$

By iteratively applying this corruption, the latent variable converges to an isotropic Gaussian distribution $z_T \sim \mathcal{N}(0, \mathbf{I})$ independent of the input.

For the reverse process, we train a denoising model parameterized by a diffusion transformer $\mu_\phi(z_t, t)$, which predicts the posterior mean and variance of the reverse transition:

$$p_\phi(z_{t-1} \mid z_t) = \mathcal{N}(z_{t-1}; \mu_\theta(z_t, t), \sigma_t^2 \mathbf{I}), \tag{2}$$

where $\sigma_t^2 = \beta_t$. After denoising, we decode the final latent $\hat{z}_0$ back into a spatial token sequence using a lightweight decoder $f^{\mathrm{dec}}$, $\hat{x}_0 = f^{\mathrm{dec}}(\hat{z}_0)$.

This formulation enables modeling the data distribution $P_{\text{data}}(\mathcal{X})$ over such trajectory space in a self-supervised manner and provides a unified framework for diverse downstream tasks, including trajectory prediction, completion, and controllable generation.

# 4 METHODOLOGY

We present an overview of the proposed GeoDiffusion, as shown in figure 1, a diffusion-based generative model designed for global-scale maritime trajectory modeling. Our method is composed of three major components: a spatial tokenization module that discretizes continuous geographic coordinates into structured grid tokens; a vessel trajectory encoder that converts spatio-temporal token sequences into latent representations; and a transformer-based latent diffusion model that progressively denoises samples in latent space.

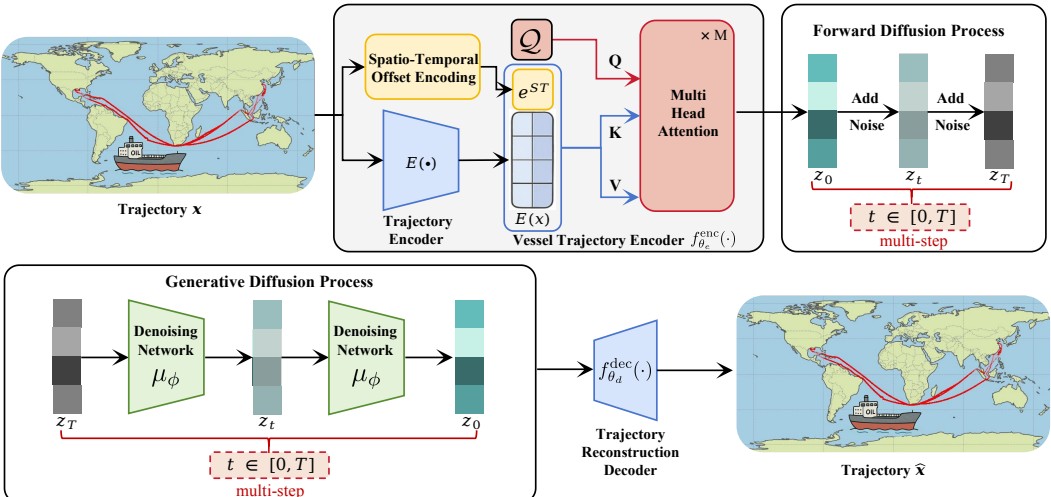

Figure 1: Overview of our proposed GeoDiffusion framework

## 4.1 VESSEL TRAJECTORY ENCODER

**Grid-based Trajectory Representation.** To enable stable training and effective modeling, we discretize the original continuous-space trajectories, represented as sequences of latitude-longitude coordinates, into a finite sequence of grid-based tokens, as shown in figure 2.

This gridification not only transforms the trajectory into a manageable, enumerable representation, but also preserves local spatial structure through the design of grid ID assignments, facilitating the learning of transition patterns between neighboring regions. Specifically, each AIS record consists of a longitude–latitude coordinate and a timestamp. We can utilize any spatial discretization function to map each geographic coordinate to a unique grid cell ID. Then, the trajectories are defined as follows:

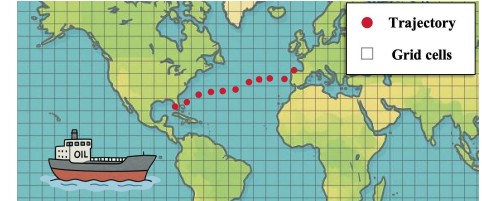

Figure 2: Grid-based representation of AIS trajectories. Each longitude–latitude coordinate is mapped to a discrete grid cell.

To effectively model the irregular sampling patterns and spatial dynamics inherent in AIS-based vessel trajectories, we propose the *Vessel Trajectory Encoder*, a neural encoding module tailored for grid-based spatio-temporal sequences. In contrast to standard positional encodings that assume uniform sampling and ignore geographic structure, our encoder incorporates a *Spatio-Temporal Offset Encoding* mechanism that explicitly captures the *relative temporal intervals* and *spatial distances* between consecutive trajectory tokens. This design enables the model to distinguish between patterns such as long-term missing segments and abrupt trajectory jumps, providing stronger inductive bias for downstream trajectory modeling tasks.

Given an input trajectory $\mathbf{x} = \{s_1, s_2, \ldots, s_N\}$, where each token $s_i = (\text{grid}_i, \tau_i)$ denotes a discrete grid cell and its associated timestamp, we first utilize a pre-trained encoder (e.g. BART (Lewis et al.,

2020)) to map variable-length $N$ tokens of $\mathbf{x}$ to a latent representation of the same length $l$ with dimension $d$, $E(\mathbf{x_0}) \in \mathbb{R}^{l \times d}$. To further enhance the representation with relational inductive bias, we incorporate a *Spatio-Temporal Offset Encoding* module that captures the relative temporal and spatial offsets between tokens.

**Spatio-Temporal Offset Encoding.** We define the *spatio-temporal offset vector* $\Delta\mathbf{x}$ for a trajectory $\mathbf{x_0}$, as the concatenation of relative spatial and temporal offsets between adjacent tokens:

$$\Delta\mathbf{x} = [\Delta s_1; \Delta s_2; \ldots; \Delta s_N] \in \mathbb{R}^{(N-1) \times 2}, \quad \text{where} \quad \Delta s_i = [\Delta g_i, \log(1 + |\tau_{i+1} - \tau_i|)] . \quad (3)$$

Here, $\Delta g_i$ denotes the difference between the grid locations of tokens $s_i$ and $s_{i+1}$, while the third term encodes the temporal gap in logarithmic scale to handle large time differences. The calculation of $\Delta g_i$ accounts for the fact that, due to the grid encoding scheme, even spatially adjacent grids may have significantly different IDs. To capture the true spatial structure, we explicitly model the relationships between grid cells. The concatenated sequence of $\{\Delta s_i\}$ forms the offset vector $\Delta\mathbf{x_0}$, which is then embedded using a two-layer MLP with ReLU (Nair & Hinton, 2010) activations to obtain the *spatio-temporal offset embedding*:

$$\mathbf{e}^{\text{ST}} = \text{MLP}_\theta(\Delta v_{ij}) \in \mathbb{R}^{(N-1) \times d}, \quad (4)$$

which captures directional and temporal context between tokens.

To integrate both semantic and structural information, we introduce a set of $k$ learnable latent queries $\mathcal{Q} \in \mathbb{R}^{k \times d}$ that perform cross-attention over the combined representation of the contextual encoder features and the offset embeddings $[E(\mathbf{x_0}); \mathbf{e}^{\text{ST}}] \in \mathbb{R}^{(l+(N-1)) \times d}$. Inspired by Alayrac et al. (2022) and Lovelace et al. (2023), we allow the latent queries to attend jointly to themselves, the frozen encoder outputs, and the spatio-temporal offset embeddings via cross-attention. Each attention block is defined as $\text{Attention}(Q, K, V) = \text{softmax}\left(\frac{QK^\top}{\sqrt{d}}\right)V$, with

$$Q = \mathcal{Q}, K = [\mathcal{Q}; E(\mathbf{x}); \mathbf{e}^{\text{ST}}], V = [\mathcal{Q}; E(\mathbf{x}); \mathbf{e}^{\text{ST}}]. \quad (5)$$

In practice, we employ a multi-head attention mechanism, and a feedforward layer follows each attention block to further refine the latent features.

We denote the entire *Vessel Trajectory Encoder* as $f_{\theta_e}^{\text{enc}}(\cdot)$, where $\theta_e$ represents the learnable parameters. Given an input trajectory $\mathbf{x}$, the encoder produces a representation $\mathbf{z} = f_{\theta_e}^{\text{enc}}(\mathbf{x})$, which is then passed to the latent diffusion process described in section 4.2.

## 4.2 LATENT DIFFUSION PROCESS FOR TRAJECTORY GENERATION

Trajectory generation in the maritime domain involves capturing both global navigational trends (e.g., long-range oceanic routes) and fine-grained local behaviors (e.g., near-port maneuvering). To flexibly model this multi-scale structure, we adopt a latent diffusion framework in which a Transformer-based denoising network serves as the core generator. The self-attention mechanism (Vaswani et al., 2017) allows each latent token to dynamically attend to both nearby and distant trajectory elements, enabling the model to adaptively focus on local details or global context as needed. This design provides a unified mechanism for handling diverse motion patterns without manually segmenting or rescaling the input, making it particularly well-suited for complex, spatio-temporally structured data such as tanker trajectories.

The Transformer-based denoising network $\mu_\phi$ is trained to reconstruct clean latent representations from their noisy counterparts. At each diffusion timestep $t$, the noisy latent $z_t$ is generated via the forward process: $z_t = \sqrt{\bar{\alpha}_t} \cdot z_0 + \sqrt{1 - \bar{\alpha}_t} \cdot \epsilon, \epsilon \sim \mathcal{N}(0, \mathbf{I})$, where $\bar{\alpha}_t = \prod_{s=1}^{t}(1 - \beta_s)$ denotes the cumulative noise factor (Ho et al., 2020; Salimans & Ho, 2022). The network $\mu_\phi$ is trained using a regression objective to predict the original clean latent $z_0$ (or the noise $\epsilon$, depending on the parameterization), with more emphasis placed on specific timesteps $t$ to improve sample quality and training efficiency.

Given an input trajectory $\mathbf{x}$, the *Vessel Trajectory Encoder* produces a representation $\mathbf{z} = f_{\theta_e}^{\text{enc}}(\mathbf{x})$. To learn the denoising process, we train a denoising network $\mu_\phi$ to recover $z$ from its noisy version using the following time-weighted regression objective:

$$\mathcal{L}(\phi) = \mathbb{E}_{t,\mathbf{z},\epsilon}[\xi_t \| \mu_\phi(\sqrt{\alpha_t}\mathbf{z} + \sqrt{1 - \alpha_t}\epsilon, t) - \mathbf{z} \|^2], \quad (6)$$

where $\epsilon \sim \mathcal{N}(0, \mathbf{I})$ is standard Gaussian noise and $\xi_t$ is a timestep-dependent weighting factor that adjusts the importance of different diffusion steps during training.

**Conditional Diffusion.** To enable control over trajectory generation, GeoDiffusion incorporates conditional information through a cross-attention mechanism inspired by Zhang et al. (2023); Lovelace et al. (2023). Specifically, we model trajectory conditions $\mathbf{c} = \{\mathbf{c}_1, \ldots, \mathbf{c}_K\} \in \mathbb{R}^{K \times d_c}$, such as departure port or destination region, as a sequence of encoded control tokens.

We append a cross-attention layer after every self-attention layer in each transformer block. The self-attention captures internal trajectory dynamics, while the cross-attention enables interaction with the external controls. Under the condition, we train the denosing network by a modified conditional regression objective:

$$\mathcal{L}(\phi) = \mathbb{E}_{t,\mathbf{z},\epsilon}[\xi_t \|\mu_\phi(\sqrt{\alpha_t}\mathbf{z} + \sqrt{1 - \alpha_t}\epsilon, t, \mathbf{c}) - \mathbf{z}\|^2]. \tag{7}$$

This design allows the model to generate trajectories consistent with high-level control signals, such as start location or destination.

**Classifier-Free Guidance (Ho & Salimans).** We apply dropout to condition tokens during training (e.g., 10%) to enable classifier-free guidance to improve sample quality. When we drop the conditioning information, we cross-attend to a learnable embedding instead of the condition.

During sampling, we enable controllable generation by interpolating predictions from conditional and unconditional models:

$$\hat{z}_t = (1 - w) \cdot \mu_\phi(z_t, t, \mathbf{c}) - w \cdot \mu_\phi(z_t, t, \emptyset), \tag{8}$$

where $w$ is the guidance scale and $\emptyset$ denotes emptyset. This allows us to flexibly balance conditioning strength without requiring a separate classifier model.

**Trajectory Reconstruction Decoder.** Following the denoising process, which transforms the standard Gaussian noise $z_T \sim \mathcal{N}(0, \mathbf{I})$ into a clean latent representation $\hat{z}_0$, we employ a lightweight decoder $f_{\theta_d}^{\text{dec}}$, parameterized by $\theta_d$, to project the latent back into the discrete spatial token space: $\hat{\mathbf{x}} = f_{\theta_d}^{\text{dec}}(\hat{z}_0)$, where each token $\hat{s}_i \in \hat{\mathbf{x}}$ denotes a predicted grid cell ID. These tokens can be subsequently mapped to geographic coordinates via a predefined grid-to-location mapping function, enabling direct visualization or downstream evaluation of the reconstructed trajectory.

**Trajectory Prediction and Imputation.** To enable trajectory prediction and imputation without requiring task-specific retraining, we design a *training-free inference strategy* based on our pretrained latent diffusion model. Given a complete trajectory represented by $N$ spatial grid tokens, we assume the first $l$ tokens are observed and accurate, while the remaining $N - l$ tokens are missing or corrupted.

Instead of retraining the model to learn a masked denoising objective, we initialize the partially observed latent as a noisy sample from an intermediate timestep $t_0$, rather than from the fully noised distribution $z_T$. Specifically, the first $L$ latent tokens are kept clean, while the remaining are corrupted with Gaussian noise scaled according to $t_0$:

$$z_{t_0}^{(i)} = \begin{cases} f_{\theta_e}^{\text{enc}}(s_i), & i \leq l \\ \sqrt{\bar{\alpha}_{t_0}} \cdot f_{\theta_e}^{\text{enc}}(s_i) + \sqrt{1 - \bar{\alpha}_{t_0}} \cdot \epsilon_i, & i > l \end{cases} \tag{9}$$

We then run the reverse diffusion process from timestep $t_0$ to $t = 0$, freezing the clean prefix at each step to ensure consistency with the known observation. Following standard practices in training-free diffusion inference, we initialize the reverse process from an intermediate timestep $t_0 = \lfloor 0.5T \rfloor$, which balances generation flexibility and trajectory fidelity. Alternative values of $t_0$ can be explored to control the strength of the reconstruction prior.

# 5 EXPERIMENTS

## 5.1 EXPERIMENT SETUP

**Pre-training Dataset.** GeoDiffusion is pre-trained on a private dataset of tanker AIS records collected from real-world maritime operations. This pre-training dataset covers AIS trajectories of over 3,500

oil tankers from January to June 2022. The AIS messages include position reports, timestamps, speed, heading, and navigational status, sampled at irregular intervals depending on transmission conditions. The vessel trajectories span global shipping routes, encompassing major ocean basins, port approaches, and intercontinental trade corridors.

**Downstream Dataset.** To evaluate the performance of GeoDiffusion, we conduct downstream experiments on two real-world AIS datasets. The first is the DAM dataset, which contains AIS trajectories of vessels operating near the Danish [1], coastline. The second is the US [2], dataset, comprising AIS records from vessels along the United States coastline. Both datasets are collected from real maritime operations and include a substantial number of tanker trajectories.

**Baselines.** We select five general-purpose state-of-the-art sequence modeling methods widely used in time-series forecasting and spatio-temporal prediction: Informer Zhou et al. (2021), Autoformer Wu et al. (2021), TimesNet Wu et al. (2023a), Crossformer Zhang & Yan (2023), and GPT-ST Li et al. (2023). These methods are adapted to our grid-based trajectory setting by embedding each token into the temporal modeling pipeline. Additionally, we include two task-specific trajectory prediction models: AISFuser Zhang et al. (2025) and AIS-Hybrid Zhu et al. (2024). Trajectory imputation: DAISTIN Magnussen et al. (2023) and PG-Diffusion Zhang et al. (2024). Route planning: DiffTraj Zhu et al. (2023), Diff-RNTraj Wei et al. (2024), ShippingMap Liu et al. (2025) and Incorporation Li & Yang (2023).

**Evaluation Metrics.** We adopt two structure-aware similarity metrics: *Path Overlap Ratio (POR)* Xu et al. (2017) and *Jaccard Similarity*, to evaluate the spatial and topological fidelity of the generated trajectories. The POR is defined as the fraction of visited grid cells in the ground-truth trajectory that are also covered by the predicted trajectory: $\text{POR} = \frac{|\text{Predicted Path} \cap \text{Ground Truth Path}|}{|\text{Ground Truth Path}|}$. In addition, we use the *Jaccard Similarity*, defined as the ratio between the intersection and union of visited grid sets: $\text{Jaccard} = \frac{|\text{Predicted Path} \cap \text{Ground Truth Path}|}{|\text{Predicted Path} \cup \text{Ground Truth Path}|}$.

**Settings.** GeoDiffusion is built on a 12-layer Transformer denoising network with a model dimension of 768, totaling about 200M parameters. More details on encoder structure, attention configuration, and training settings are provided in Appendix A.1.

## 5.2 DOWNSTREAM TASKS

**Trajectory Prediction.** Forecasting a vessel's future positions in continuous geographic space based on its movement history supports fine-grained maritime navigation and safety applications. It enables high-resolution path planning, real-time collision avoidance, and anticipatory traffic coordination. The task involves modeling complex temporal and spatial dependencies while dealing with nonlinearity and irregular sampling in trajectory data.

**Trajectory Imputation.** Completing partially observed vessel trajectories by inferring missing segments is crucial for data quality assurance in AIS-based analytics. This task addresses AIS signal loss due to environmental or technical factors and recovers plausible intermediate paths. Effective imputation improves downstream reliability in tasks such as behavior classification, anomaly detection, and movement pattern mining.

**Route Planning.** Generating feasible trajectories between specified origin and destination locations on the maritime grid supports intelligent decision-making in ship routing, logistics scheduling, and fuel-efficient operations. The task requires the model to synthesize globally consistent and context-aware trajectories that align with navigational norms and physical movement constraints. It reflects the model's ability to generalize beyond observed paths and perform conditional trajectory generation.

## 5.3 MAIN RESULTS

**Overview.** As a foundation model for maritime mobility, GeoDiffusion achieves state-of-the-art performance across a range of trajectory prediction, imputation, and route planning tasks, demonstrating strong generalization ability and robustness to spatial and temporal uncertainty. Detailed comparisons are shown in the following paragraphs. In all tables, we mark the best results of POR in bold, and the

---

[1] https://www.dma.dk/safety-at-sea/navigational-information/ais-data
[2] https://hub.marinecadastre.gov/pages/vesseltraffic

best results of Jaccard in underlined.

Table 1: Trajectory prediction results on three datasets under different forecasting lengths (shorter input-output windows). We report POR (↑) and Jaccard Similarity (↑). Higher is better.

| Tasks | | GeoDiffusion | | Informer | | Autoformer | | TimesNet | | Crossformer | | GPT-ST | | AISFuser | | AIS-Hybrid | |
|---|---|---|---|---|---|---|---|---|---|---|---|---|---|---|---|---|---|
| | | POR | Jaccard | POR | Jaccard | POR | Jaccard | POR | Jaccard | POR | Jaccard | POR | Jaccard | POR | Jaccard | POR | Jaccard |
| Private Dataset | 10-5 | 0.790 | 0.770 | 0.840 | 0.805 | 0.818 | 0.785 | **0.856** | 0.820 | 0.752 | 0.728 | 0.790 | 0.760 | 0.810 | 0.785 | 0.828 | 0.798 |
| | 17-8 | **0.818** | 0.785 | 0.812 | 0.780 | *0.732* | 0.710 | 0.742 | 0.720 | 0.758 | 0.735 | 0.786 | 0.750 | 0.736 | 0.710 | 0.738 | 0.698 |
| | 23-12 | **0.852** | 0.820 | 0.760 | 0.735 | 0.738 | 0.705 | 0.748 | 0.713 | 0.778 | 0.740 | 0.718 | 0.680 | 0.722 | 0.692 | 0.703 | 0.670 |
| | 30-15 | **0.830** | 0.735 | 0.686 | 0.657 | 0.700 | 0.678 | 0.807 | 0.775 | 0.829 | 0.795 | 0.770 | 0.800 | 0.765 | 0.742 | 0.736 | 0.710 |
| | 40-20 | **0.825** | 0.800 | 0.736 | 0.700 | 0.742 | 0.720 | 0.757 | 0.728 | 0.772 | 0.750 | 0.723 | 0.702 | 0.724 | 0.702 | 0.698 | 0.664 |
| DAM Dataset | 10-5 | **0.922** | 0.890 | 0.905 | 0.872 | 0.894 | 0.864 | 0.917 | 0.877 | 0.880 | 0.852 | 0.905 | 0.870 | 0.892 | 0.858 | 0.888 | 0.850 |
| | 17-8 | **0.902** | 0.865 | 0.878 | 0.845 | 0.866 | 0.835 | 0.870 | 0.840 | 0.865 | 0.828 | 0.847 | 0.810 | 0.860 | 0.830 | 0.843 | 0.805 |
| | 23-12 | **0.890** | 0.852 | 0.861 | 0.825 | 0.845 | 0.812 | 0.858 | 0.822 | 0.867 | 0.835 | 0.845 | 0.812 | 0.852 | 0.820 | 0.833 | 0.792 |
| | 30-15 | **0.865** | 0.828 | 0.840 | 0.805 | 0.834 | 0.800 | 0.847 | 0.812 | 0.842 | 0.805 | 0.828 | 0.790 | 0.834 | 0.798 | 0.815 | 0.780 |
| | 40-20 | **0.848** | 0.812 | 0.815 | 0.780 | 0.808 | 0.775 | 0.820 | 0.785 | 0.822 | 0.790 | 0.805 | 0.770 | 0.800 | 0.765 | 0.783 | 0.748 |
| U.S. Coast Dataset | 10-5 | **0.905** | 0.870 | 0.875 | 0.840 | 0.855 | 0.825 | 0.868 | 0.835 | 0.865 | 0.832 | 0.880 | 0.850 | 0.865 | 0.830 | 0.850 | 0.820 |
| | 17-8 | **0.875** | 0.840 | 0.850 | 0.815 | 0.835 | 0.800 | 0.845 | 0.810 | 0.842 | 0.805 | 0.835 | 0.798 | 0.820 | 0.785 | 0.813 | 0.772 |
| | 23-12 | **0.860** | 0.825 | 0.835 | 0.800 | 0.820 | 0.785 | 0.830 | 0.795 | 0.818 | 0.785 | 0.815 | 0.780 | 0.803 | 0.770 | 0.795 | 0.755 |
| | 30-15 | **0.840** | 0.805 | 0.815 | 0.780 | 0.805 | 0.770 | 0.812 | 0.775 | 0.810 | 0.772 | 0.803 | 0.765 | 0.792 | 0.758 | 0.785 | 0.748 |
| | 40-20 | **0.815** | 0.785 | 0.788 | 0.755 | 0.778 | 0.745 | 0.787 | 0.752 | 0.782 | 0.748 | 0.775 | 0.742 | 0.765 | 0.735 | 0.757 | 0.722 |

**Trajectory Prediction.** Table 1 (top) reports the performance of GeoDiffusion and baseline methods across three datasets, Private, DAM, and U.S. Coast, under multiple forecasting horizons (e.g., 10-5, 23-12, 40-20), where each setting denotes the number of observed days and future days to predict. GeoDiffusion consistently achieves the highest or second-highest scores in both POR and Jaccard Similarity across almost all settings and datasets, highlighting its strength in modeling long-range dependencies and handling irregular sampling. Notably, GeoDiffusion maintains superior performance even on the large and diverse Private dataset, showcasing its scalability. Compared to strong time-series baselines like Informer (Zhou et al., 2021), Autoformer (Wu et al., 2021), and TimesNet (Wu et al., 2023a), GeoDiffusion better captures spatio-temporal dynamics and generates more coherent and realistic trajectories. Furthermore, domain-specific baselines such as AISFuser (Zhang et al., 2025) and AIS-Hybrid (Zhu et al., 2024) are outperformed in nearly all configurations, reinforcing the effectiveness of our approach.

Table 2: Trajectory imputation results on three datasets under different missing patterns (shorter input-output windows). Each setting indicates the total duration and randomly missing days. We report POR (↑) and Jaccard (↑).

| Tasks | | GeoDiffusion | | Informer | | Autoformer | | TimesNet | | Crossformer | | GPT-ST | | DAISTIN | | PG-Diffusion | |
|---|---|---|---|---|---|---|---|---|---|---|---|---|---|---|---|---|---|
| | | POR | Jaccard | POR | Jaccard | POR | Jaccard | POR | Jaccard | POR | Jaccard | POR | Jaccard | POR | Jaccard | POR | Jaccard |
| Private Dataset | 10-5 | 0.710 | 0.685 | 0.749 | 0.716 | **0.770** | 0.740 | 0.630 | 0.610 | 0.647 | 0.615 | 0.637 | 0.605 | 0.738 | 0.710 | 0.740 | 0.705 |
| | 17-8 | **0.750** | 0.725 | 0.678 | 0.650 | 0.650 | 0.620 | 0.683 | 0.653 | 0.745 | 0.710 | 0.705 | 0.668 | 0.704 | 0.668 | 0.695 | 0.670 |
| | 23-12 | **0.748** | 0.713 | 0.687 | 0.665 | 0.660 | 0.625 | 0.739 | 0.703 | 0.698 | 0.665 | 0.680 | 0.648 | 0.675 | 0.644 | 0.665 | 0.630 |
| | 30-15 | **0.742** | 0.710 | 0.644 | 0.610 | 0.730 | 0.700 | 0.628 | 0.595 | 0.707 | 0.673 | 0.655 | 0.618 | 0.737 | 0.715 | 0.657 | 0.625 |
| | 40-20 | **0.790** | 0.760 | 0.630 | 0.605 | 0.705 | 0.668 | 0.655 | 0.625 | 0.772 | 0.740 | 0.670 | 0.635 | 0.645 | 0.618 | 0.717 | 0.690 |
| DAM Dataset | 10-5 | **0.835** | 0.812 | 0.823 | 0.792 | 0.842 | 0.808 | 0.795 | 0.765 | 0.810 | 0.783 | 0.792 | 0.760 | 0.795 | 0.765 | 0.808 | 0.775 |
| | 17-8 | **0.805** | 0.772 | 0.789 | 0.755 | 0.771 | 0.738 | 0.790 | 0.753 | 0.785 | 0.750 | 0.772 | 0.738 | 0.778 | 0.743 | 0.794 | 0.760 |
| | 23-12 | **0.802** | 0.765 | 0.772 | 0.740 | 0.757 | 0.723 | 0.770 | 0.738 | 0.764 | 0.733 | 0.758 | 0.725 | 0.747 | 0.712 | 0.755 | 0.720 |
| | 30-15 | **0.815** | 0.785 | 0.761 | 0.725 | 0.792 | 0.758 | 0.755 | 0.720 | 0.782 | 0.750 | 0.758 | 0.723 | 0.772 | 0.740 | 0.777 | 0.743 |
| | 40-20 | **0.828** | 0.800 | 0.750 | 0.715 | 0.775 | 0.740 | 0.750 | 0.715 | 0.808 | 0.775 | 0.762 | 0.728 | 0.755 | 0.722 | 0.773 | 0.738 |
| U.S. Coast Dataset | 10-5 | **0.832** | 0.798 | 0.818 | 0.785 | 0.832 | 0.798 | 0.790 | 0.758 | 0.805 | 0.772 | 0.787 | 0.755 | 0.792 | 0.760 | 0.802 | 0.770 |
| | 17-8 | **0.808** | 0.775 | 0.785 | 0.750 | 0.762 | 0.728 | 0.787 | 0.753 | 0.782 | 0.747 | 0.770 | 0.735 | 0.777 | 0.743 | 0.790 | 0.757 |
| | 23-12 | **0.795** | 0.760 | 0.765 | 0.730 | 0.750 | 0.715 | 0.764 | 0.730 | 0.757 | 0.723 | 0.750 | 0.715 | 0.740 | 0.705 | 0.748 | 0.712 |
| | 30-15 | **0.798** | 0.765 | 0.745 | 0.710 | 0.775 | 0.742 | 0.740 | 0.705 | 0.770 | 0.738 | 0.742 | 0.708 | 0.758 | 0.725 | 0.762 | 0.730 |
| | 40-20 | **0.817** | 0.785 | 0.735 | 0.700 | 0.765 | 0.730 | 0.740 | 0.705 | 0.795 | 0.765 | 0.748 | 0.712 | 0.742 | 0.710 | 0.758 | 0.725 |

**Trajectory Imputation.** Table 2 presents the results of trajectory imputation under various missing patterns. Each setting (e.g., 17-8, 23-12) denotes the total duration and the number of randomly missing days. GeoDiffusion outperforms all baselines across most

Table 3: Route planning results on three datasets with two types of point pairs (between ports or randomly selected points). We report POR (↑) and Jaccard Similarity (↑). Higher is better.

| Tasks | | GeoDiffusion | | Difftraj | | Diff-RNTraj | | ShippingMap | | Incorporation | |
|---|---|---|---|---|---|---|---|---|---|---|---|
| | | POR | Jaccard | POR | Jaccard | POR | Jaccard | POR | Jaccard | POR | Jaccard |
| Private Dataset | ports | **0.8688** | 0.8377 | 0.7752 | 0.8213 | 0.7532 | 0.8017 | 0.7634 | 0.7288 | 0.7935 | 0.7547 |
| | random | **0.8406** | 0.8188 | 0.7518 | 0.7150 | 0.7565 | 0.7355 | 0.7720 | 0.7438 | 0.7869 | 0.7651 |
| DAM Dataset | ports | **0.9375** | 0.9033 | 0.9213 | 0.8871 | 0.9099 | 0.8780 | 0.9327 | 0.8916 | 0.8975 | 0.8685 |
| | random | **0.8594** | 0.8263 | 0.8265 | 0.7916 | 0.8196 | 0.7849 | 0.8297 | 0.7948 | 0.8312 | 0.8015 |
| U.S. Coast Dataset | ports | **0.9188** | 0.8823 | 0.8884 | 0.8526 | 0.8667 | 0.8371 | 0.8811 | 0.8495 | 0.8790 | 0.8471 |
| | random | **0.8250** | 0.7910 | 0.7950 | 0.7591 | 0.7846 | 0.7486 | *0.7928* | *0.7554* | 0.7874 | 0.7522 |

settings in both POR and Jaccard metrics, particularly under high missing rates (e.g., 30-15, 40-20), where the generative capability of the diffusion model is most needed. While DAISTIN (Magnussen et al., 2023) and PG-Diffusion (Zhang et al., 2024) offer tailored designs for interpolation, GeoDiffusion surpasses them with a simpler, training-free conditional generation mechanism that adapts to different missing patterns without retraining. This demonstrates its strong adaptability and robustness to missing data, a key characteristic of real-world AIS scenarios. The consistent

performance across all three datasets again confirms GeoDiffusion's generalization capability and suitability as a general-purpose maritime foundation model.

**Route Planning.** To assess GeoDiffusion's effectiveness in real-world maritime planning, we evaluate its performance on trajectory generation between two given locations, considering both (1) port-to-port and (2) randomly selected point pairs, simulating both common and rare routing demands. As shown in Table 3, across the Private, DAM, and U.S. Coast datasets, GeoDiffusion consistently achieves the best overall performance in terms of POR and Jaccard Similarity. Notably, its advantage is more evident in the random-point setting, highlighting its strong generalization and ability to model diverse, long-range routes. Compared to diffusion-based baselines like Difftraj and Diff-RNTraj, which struggle with random paths, and task-specific models like ShippingMap and Incorporation, which lack robustness across datasets, GeoDiffusion delivers accurate and adaptable trajectory generation across all scenarios.

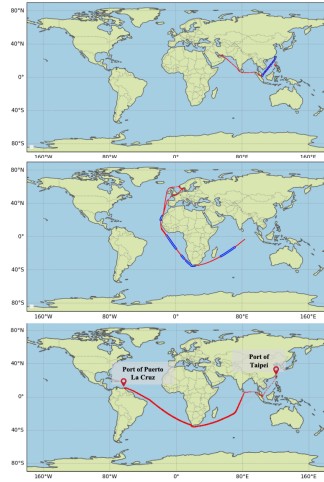

Figure 3: Visualization.

**Visualization.** To further evaluate the modeling capacity of GeoDiffusion, we visualize its outputs on three representative downstream tasks in Figure 3. The top panel illustrates trajectory prediction, where the model generates coherent continuations of observed routes that align with real-world navigation, capturing both constrained passages in narrow seas and diverse options in open waters. The middle panel presents trajectory imputation under missing data, showing that GeoDiffusion produces smooth and globally consistent completions that remain plausible over long distances, underscoring its robustness to data sparsity in real AIS scenarios. The bottom panel depicts route planning between the Port of Taipei and the Port of Puerto La Cruz, where the model generates feasible maritime paths that respect geographic constraints and avoid unrealistic shortcuts across land. Together, these visualizations demonstrate GeoDiffusion's ability to capture both global regularities and regional uncertainties across prediction, imputation, and planning tasks, highlighting its versatility and applicability in complex maritime environments. More visualization results of the three sub-tasks are provided in the Appendix A.5.

**Ablation Study** To evaluate the necessity of key components in GeoDiffusion, we conduct ablation experiments on the Private Dataset under the 60-32 setting. Results in Table 4 show that removing the Spatio-Temporal Offset Encoding (STOE) leads to a performance drop of 3.4% in POR and 3.9% in Jaccard, confirming the importance of explicitly modeling relative offsets. Eliminating the

Table 4: Key ablations on the Private Dataset (60-32 setting). We report Path Overlap Ratio (POR↑) and Jaccard Similarity (↑).

| Setting | POR | Jaccard |
|---|---|---|
| Full GeoDiffusion | **0.8463** | **0.7491** |
| w/o Spatio-Temporal Offset Encoding | 0.8122 | 0.7100 |
| w/o Trajectory Constraints | 0.8071 | 0.7103 |
| Auto-Regressive Generation | 0.7896 | 0.6932 |

trajectory constraint mechanism destabilizes predictions and degrades both metrics by more than 5%, while replacing diffusion with an auto-regressive objective yields competitive short-term forecasting but significantly worse long-horizon results due to error accumulation. Overall, these results highlight that each component is indispensable for stable and accurate trajectory generation, while further ablation analyses are reported in Appendix A.2.

## 6 CONCLUSION

We introduced GeoDiffusion, a diffusion-based foundation model for maritime trajectory modeling under irregular and noisy spatio-temporal conditions. By combining a Spatio-Temporal Offset Encoding, a Transformer-based denoising network, and a training-free conditional inference strategy, GeoDiffusion unifies forecasting, imputation, and planning within a single pretrained framework. Experiments on large-scale AIS datasets demonstrate state-of-the-art performance across all tasks, while visualizations highlight the model's ability to capture both global regularities and regional uncertainties. These results establish GeoDiffusion as a scalable and versatile solution for maritime mobility, and a general blueprint for irregular spatio-temporal foundation models.

One limitation of our current framework is that it exclusively focuses on tanker trajectories, without incorporating other types of vessels such as cargo ships or passenger liners, which may exhibit distinct movement patterns and operational behaviors.

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
