# A APPENDIX

## A.1 EXPERIMENT SETTINGS IN DETAIL

**Model Configurations.** The denoising network in GeoDiffusion is implemented as a 12-layer pre-LayerNorm Transformer with a hidden dimension of 768. The Spatio-Temporal Trajectory Encoder is composed of 2 layers of cross-attention blocks, each equipped with multi-head attention using 8 attention heads. Each cross-attention layer is followed by a feed-forward network with an inner dimension of 1024, residual connections, and LayerNorm applied before both the attention and feed-forward modules. In total, GeoDiffusion contains approximately 200M parameters.

**Training Details.** All models are trained using AdamW optimizer with an initial learning rate of 1e-4 and linear warm-up for the first 10k steps. The diffusion process is discretized into 1000 steps with cosine noise scheduling. Batch size is set to 64, and dropout rate is 0.1. Early stopping is applied based on validation loss.

**Evaluation Protocol.** Models are evaluated under multiple trajectory forecasting and imputation settings. For each case, we report Path Overlap Ratio (POR) and Jaccard Similarity as main metrics. Unless otherwise specified, all reported results are averaged across three independent runs.

## A.2 ADDITIONAL ABLATION RESULTS

For completeness, we include additional ablations that further validate GeoDiffusion's design. As shown in Table 5, replacing the Transformer-based denoising network with a convolutional alternative reduces the ability to capture long-range dependencies, leading to over 5% degradation on both metrics. Removing the latent query cross-attention module also weakens semantic awareness, resulting in a moderate performance drop. These results support the necessity of multi-head attention and hybrid representations in strengthening trajectory generation.

Table 5: Extended ablation results on the trajectory prediction task (Private Dataset, 60-32 setting).

| Setting | POR | Jaccard |
|---|---|---|
| w/o Transformer (ConvNet Denoiser) | 0.8025 | 0.7073 |
| w/o Latent Query Cross-Attention | 0.8250 | 0.7234 |

## A.3 GEODIFFUSION AND BASELINE IMPLEMENTATION DETAILS

### A.3.1 GEODIFFUSION IMPLEMENTATION

We implement GeoDiffusion using PyTorch and train it on a single NVIDIA A40 GPU with 48 GB of memory. The model consists of three main components: a Vessel Trajectory Encoder, a Transformer-based denoising network, and a trajectory reconstruction decoder. The encoder processes AIS trajectories represented as sequences of spatio-temporal tokens $\mathbf{x} = \{s_1, s_2, \ldots, s_N\}$, where each token $s_i = (\text{grid}_i, \tau_i)$ combines a grid cell ID and timestamp. Grid tokens are embedded into 512-dimensional vectors, and we introduce a Spatio-Temporal Offset Encoding mechanism to model relative spatial and temporal differences between adjacent tokens. The resulting features are fused with frozen BART encoder outputs via cross-attention, yielding a latent representation $\mathbf{z}_0 \in \mathbb{R}^{k \times 512}$, with $k = 16$. The denoising network is a 12-layer Transformer trained using the DDPM framework with a linear noise schedule over $T = 1000$ steps. We use a standard noise-prediction objective with a time-weighting term. After denoising, the trajectory decoder transforms the final latent $\hat{\mathbf{z}}_0$ into grid token predictions via an MLP and softmax, which are mapped back to geographic coordinates. We train the model with a batch size of 128, learning rate of 1e-4, AdamW optimizer, weight decay of 0.01, and dropout of 0.1. Gradient clipping (1.0) and linear learning rate warm-up over the first 10k steps are applied. Task-specific conditioning mechanisms are used: for prediction, the known prefix is fixed; for imputation, masked tokens are denoised while observed ones remain fixed; for planning, only endpoints are fixed while intermediate tokens are generated. Training on the private dataset takes roughly 2 days, and inference runs at under 50 ms per trajectory using 50 denoising steps. All results are averaged over 5 random seeds.

### A.3.2 BASELINES

To evaluate the effectiveness of our method across different tasks, we compare GeoDiffusion with a comprehensive set of baselines, categorized according to the nature of each task.

*Trajectory Prediction.* We select five general-purpose state-of-the-art sequence modeling methods widely used in time-series forecasting and spatio-temporal prediction: Informer Zhou et al. (2021), Autoformer Wu et al. (2021), TimesNet Wu et al. (2023a), Crossformer Zhang & Yan (2023), and GPT-ST Li et al. (2023). These methods are adapted to our grid-based trajectory setting by embedding each token into the temporal modeling pipeline. Additionally, we include two task-specific trajectory prediction models: AISFuser Zhang et al. (2025) and AIS-Hybrid Zhu et al. (2024).

*Trajectory Imputation.* We use the same five general-purpose baselines for sequence modeling as in the prediction task. To account for methods explicitly designed for trajectory completion, we further compare with DAISTIN Magnussen et al. (2023) and PG-Diffusion Zhang et al. (2024).

*Route Planning.* Due to the structured and goal-directed nature of the route planning task, we select two diffusion-based generative models as general baselines: DiffTraj Zhu et al. (2023) and Diff-RNTraj Wei et al. (2024). We also include two maritime domain planning models: ShippingMap Liu et al. (2025) and Incorporation Li & Yang (2023).

### A.4 BASELINE IMPLEMENTATION

**Informer** Zhou et al. (2021) employs ProbSparse self-attention to improve the efficiency of long sequence modeling and has been widely adopted for time series forecasting tasks. To implement Informer, we utilize their open-source code, available at https://github.com/zhouhaoyi/Informer2020. Following the original configuration, we use the encoder-decoder variant with 3 encoder layers and 2 decoder layers, and set the embedding dimension to 512. The total number of parameters is about 18.36M. We set the number of training epochs to 50, the learning rate to 0.001, and the batch size to 128. Input and prediction sequence lengths are aligned with our grid-based trajectory forecasting protocol, and grid IDs are embedded using a shared trainable embedding layer.

**Autoformer** Wu et al. (2021) designs a decomposition-based forecasting framework that captures temporal trends and seasonalities. We implement Autoformer using their official repository, available at https://github.com/thuml/Autoformer. We adopt the 3-layer encoder-only variant with an embedding dimension of 512, which shows robust performance in the original paper. The total model size is 19.45M. We set the number of training epochs to 50, the learning rate to 0.001, and the batch size to 128. During preprocessing, each grid token is mapped to a learned embedding and fed into the temporal encoder for sequence-to-sequence forecasting.

**TimesNet** Wu et al. (2023a) models time series data in a 2D variation space to capture local and global temporal dynamics. We follow their official implementation at https://github.com/thuml/TimesNet. Based on the default configuration, we use 3 convolutional blocks with an embedding size of 512 and keep all other hyperparameters unchanged. The total model size is 20.91M. We train the model for 50 epochs with a learning rate of 0.001 and a batch size of 128. We reshape the trajectory grid token sequence as a univariate time series and adapt TimesNet to handle both forecasting and imputation by applying masking strategies consistent with our task setup.

**Crossformer** Zhang & Yan (2023) introduces cross-dimension dependency modeling for multivariate time series forecasting. We implement Crossformer using the official codebase at https://github.com/Thinklab-SJTU/Crossformer. Following the base configuration, we use 2 cross blocks with an embedding dimension of 512 and maintain all default hyperparameters. The total parameter size is 17.02M. The model is trained for 50 epochs with a learning rate of 0.001 and a batch size of 128. For input adaptation, each trajectory is treated as a multivariate sequence by stacking spatial and temporal token embeddings along the feature axis, enabling Crossformer to model both spatial locality and temporal correlation.

**GPT-ST** Li et al. (2023) proposes a generative pretraining framework for spatio-temporal graph neural networks, utilizing masked token prediction to capture complex spatial-temporal dependencies. To implement GPT-ST in our trajectory modeling setting, we adopt their official codebase, available at https://github.com/HKUDS/GPT-ST. Since GPT-ST was originally designed for traffic flow data over road networks, we adapt it to our grid-based maritime trajectories by constructing spatial graphs over

grid cells. Specifically, we define undirected edges between adjacent grid cells based on 8-neighbor spatial adjacency, and construct trajectory-induced temporal graphs where each node corresponds to a grid cell at a specific time step.

Following the encoder-only variant of GPT-ST, we treat each trajectory as a sequence of graph node tokens and apply masked modeling to train the model to recover randomly masked grid IDs. For trajectory prediction, the model is trained to predict future tokens given partial input, and for imputation, we randomly mask segments and train the model to fill in the missing positions. We use a 3-layer transformer encoder with hidden size 512, consistent with the original implementation. The total model size is 21.73M. We train GPT-ST for 50 epochs using the Adam optimizer with a learning rate of 0.001 and a batch size of 128. Masking ratios and sampling strategies follow the default settings from the official implementation.

**AISFuser** Zhang et al. (2025) is a state-of-the-art transformer-based framework that models maritime graphical representations alongside multi-modal temporal dynamics for vessel trajectory prediction. It integrates both spatial constraints from a maritime traffic network and temporal heterogeneity via a self-supervised learning (SSL) module. To implement AISFuser, we follow the original settings and utilize the publicly available AIS dataset provided by the Danish Maritime Authority. The encoder includes 6 transformer layers with 8 attention heads, and the AIS data is discretized with a spatial resolution of 0.01° for LAT/LON and embedded into a 256-dimensional latent space. Maritime graphical structures (e.g., ship lanes and landscape) within a 50 km radius are encoded using an attention-based polyline graph module. We train the model for 50 epochs using the Adam optimizer with a learning rate of 0.001 and a batch size of 64. Predictions are made over a variable forecast horizon of up to 10 hours. The SSL loss weight is set to 0.1. We use the official code provided by the authors and evaluate the model using MAE and RMSE metrics on the discretized grid-based trajectory outputs.

**AIS-Hybrid** Zhu et al. (2024) integrates a seamanship-informed encounter classification module with a two-branch hybrid predictor combining kinematics-based and neural network (NN)-based models. To implement AIS-Hybrid, we follow the methodology described in the original paper and reproduce the two-stage pipeline: an encounter classification phase based on TCPA and COG distributions, followed by a trajectory predictor. For prediction, the NN branch uses a single-layer LSTM encoder-decoder with 128 hidden units and a fully connected output layer, trained for 400–1000 epochs depending on encounter type. The kinematics branch uses historical AIS data and selects a best-matching trajectory via DTW-based similarity retrieval. The final prediction is obtained via an adaptive weighted fusion of the two branches based on real-time error feedback. We discretize AIS positions into grid IDs and resample the data to a 1-second interval to match our grid-based prediction format. Evaluation follows the original paper's metrics, using Mean Prediction Error (MPE) computed over a 3–9 minute prediction horizon. The model is trained and tested on the Oslofjord AIS dataset provided in the original study.

**DAISTIN** Magnussen et al. (2023) is a data-driven AIS trajectory interpolation method that reconstructs vessel trajectories by searching shortest paths over a directed graph constructed from massive historical AIS data. To implement DAISTIN, we follow the original configuration described in the paper and reproduce the full pipeline, including digraph construction, interpolation via A* pathfinding, and postprocessing. The graph is built from AIS trajectory segments after geometric sampling using a spatial density-aware strategy with 50 quantiles and a sampling rate of $910^6$ data points. Each AIS node encodes latitude, longitude, and heading, and edge connections are determined by a spatial neighborhood radius of 20 km and a direction threshold of 45 degrees. During inference, signal gaps exceeding this radius are interpolated via shortest paths, and Douglas-Peucker postprocessing is applied to reduce zigzag noise. For consistency with our grid-based formulation, interpolated positions are mapped to discrete grid IDs. We use the optimal hyperparameter settings reported in the original paper and evaluate DAISTIN using the same SPD and HD metrics. The model is tested on the same IHS Markit tanker AIS dataset used in the original study, filtered to match our imputation protocol with at least 20% missing samples in each trajectory.

**PG-Diffusion** Zhang et al. (2024) introduces a physics-guided diffusion probabilistic framework designed specifically for long-term vessel trajectory imputation. To implement PG-Diffusion, we follow the original setting and use the official AIS dataset provided by the Danish Maritime Authority, filtered for cargo and tanker vessels. We adopt the original architecture, which includes a pre-trained variational trajectory embedding module (VRNN), a transformer encoder for modeling observed

trajectories, and a Traj-UNet denoising backbone with physics-based constraints. The transformer encoder is configured with 3 layers, 4 attention heads, and 128 hidden units. The number of diffusion time steps is set to 500. During training, we incorporate sinusoidal timestep embeddings and vessel/time ID embeddings into the noise prediction module. The physics-guided discriminator, active only during training, enforces kinematic consistency between consecutive imputed points via Earth motion equations. We set the discriminator step to 20, and the hyperparameters for physics loss terms $\beta_1$ and $\beta_2$ are set to 0.2 and 0.1, respectively. The model is trained for 50 epochs using the Adam optimizer with a learning rate of $1 \times 10^{-4}$ and a batch size of 512. For compatibility with our grid-based formulation, all imputed GPS coordinates are mapped to discrete grid IDs before evaluation.

**DiffTraj** Zhu et al. (2023) introduces a spatial-temporal diffusion probabilistic model for realistic GPS trajectory generation. To implement DiffTraj, we utilize the official PyTorch implementation provided by the authors, available at https://github.com/Yasoz/DiffTraj. The model includes a forward Gaussian noise process and a reverse denoising process, parameterized by the Traj-UNet architecture with 4 downsampling and upsampling blocks and 2 ResNet blocks per stage. Conditional generation is enabled using a Wide & Deep module that embeds contextual inputs such as departure region and time. To adapt the model for our Route Planning task, we explicitly provide start and end grid tokens as part of the conditional input. The model is trained using a 500-step diffusion schedule with a batch size of 1024 and learning rate of $1 \times 10^{-4}$, following the original setting. For evaluation, generated trajectories are decoded into discrete grid sequences and compared against ground truth using Grid Edit Distance (GED) and Path Overlap Ratio (POR) metrics.

**Diff-RNTraj** Wei et al. (2024) proposes a structure-aware diffusion model designed for road network-constrained trajectory generation. To implement Diff-RNTraj, we follow the authors' official implementation available at https://github.com/wtl52656/Diff-RNTraj. The model consists of three main components: a hybrid RNTraj vectorization module that embeds discrete road segments (via pre-trained Node2vec representations) and continuous movement ratios into a continuous space; a continuous diffusion model trained over 500 denoising steps; and a RNTraj decoder that reconstructs road segment sequences and locations from the generated latent representations. For our Route Planning task, we adapt the conditional sampling pipeline by inserting the specified origin and destination road segments into the diffusion process as soft positional anchors. The denoising backbone is built with 10 residual dilation convolution layers (RDCLs) with kernel size 3 and 512 channels, and we use a batch size of 256, a learning rate of 0.001, and the Adam optimizer. All generated trajectories are decoded to grid-based sequences and evaluated using Grid Edit Distance (GED) and Path Overlap Ratio (POR) against ground-truth routes.

**ShippingMap** Liu et al. (2025) introduces a hybrid data-driven and model-based framework for global vessel route planning by constructing a static grid-based maritime network from large-scale AIS data. To implement ShippingMap, we follow the original four-step process: AIS trajectory segmentation using spatiotemporal thresholds, clustering of berthing and waypoint areas via CKBA-DBSCAN, maritime shipping network construction using historical trajectory arcs, and double-layer A*-based route search. Specifically, we partition the global water surface into a 1800900 grid system (0.2° resolution) and construct a directed grid network based on historical navigation. Waypoints are detected via the PELT change point algorithm and grouped using directional-aware DBSCAN clustering. Given a source and destination port, we map them to corresponding grid cells and apply A* search on the combined shipping and grid network to retrieve the optimal route. The code is implemented in Python and evaluated on the global 2018 AIS dataset of dry bulk carriers. To align with our grid-based representation, all planned routes are discretized into sequences of grid tokens. We adopt Grid Edit Distance (GED) and Path Overlap Ratio (POR) as evaluation metrics for comparing the planned routes with ground-truth trajectories.

**AIS-MASS** Li & Yang (2023) proposes an unsupervised route planning framework for Maritime Autonomous Surface Ships (MASS) based on historical AIS data. To implement AIS-MASS, we follow the original methodology, which consists of two key components: movement pattern extraction via Automatic and Adaptive Dynamic Time Warping (AADTW) and unsupervised clustering using the Spectral Clustering with Affinity Feature (SCAF) algorithm. Movement patterns are mined separately for different ship types (e.g., tankers and cargo vessels) and are used to construct representative route libraries. For each planning instance, given a source and destination location, the algorithm identifies the nearest feature centers in the cluster space and retrieves the corresponding optimal pattern route. The system is fully data-driven and does not require any human-labeled samples. In our adaptation

to the grid-based Route Planning setting, we map all AIS coordinates to discrete grid tokens and apply the AADTW+SCAF pipeline to extract cluster-wise centroid routes. During inference, we use a nearest-cluster search based on source-destination grid alignment and evaluate the retrieved routes using Grid Edit Distance (GED) and Path Overlap Ratio (POR). All hyperparameters (e.g., clustering affinity weights and DTW window size) follow the settings recommended in the original implementation.

**Evaluation Metrics.** We adopt two structure-aware similarity metrics: *Path Overlap Ratio (POR)* and *Jaccard Similarity*, to evaluate the spatial and topological fidelity of the generated trajectories. The *Path Overlap Ratio (POR)* is defined as the fraction of visited grid cells in the ground-truth trajectory that are also covered by the predicted trajectory: $\text{POR} = \frac{|\text{Predicted Path} \cap \text{Ground Truth Path}|}{|\text{Ground Truth Path}|}$. This metric emphasizes *coverage completeness*—how well the generated route captures the essential waypoints of the true trajectory. It is particularly suitable for maritime path generation where key navigation points (e.g., turn regions, straits) must be preserved for operational validity. In addition, we use the *Jaccard Similarity*, defined as the ratio between the intersection and union of visited grid sets: $\text{Jaccard} = \frac{|\text{Predicted Path} \cap \text{Ground Truth Path}|}{|\text{Predicted Path} \cup \text{Ground Truth Path}|}$. Unlike POR, the Jaccard index penalizes both under-coverage and over-coverage, thus providing a balanced assessment of *path precision* and *path recall*. Together, these two metrics capture both *coverage sufficiency* and *topological compactness*, offering a comprehensive view of performance in grid-based maritime route planning.

## A.5 MORE VISUALIZATION RESULTS

To further illustrate GeoDiffusion's prediction and imputation capacity, we present representative visualization examples from all experimental settings. In each case, the red trajectory denotes observed historical AIS data, and the blue trajectory indicates the predicted continuation. For each setting, we randomly select 8 representative cases for display.

**15 days imputation.** This setting corresponds to imputing vessel trajectories with 15 days of observed history and subsequent missing intervals to be filled. As shown in Figure 5, GeoDiffusion is able to reconstruct plausible continuations that align with global shipping routes while adapting to diverse missing patterns.

**15 days prediction.** In this case, 10 days of historical trajectory are given and the next 5 days are forecasted. Figure 6 shows 8 examples, where the model robustly captures short-term continuation across different oceanic regions.

**25 days imputation.** Here, GeoDiffusion imputes missing segments after 25 days of observed history. Figure 7 illustrates the ability to maintain both local consistency and long-range realism under higher missing ratios.

**25 days prediction.** Using 20 days of observed trajectory, the model predicts the following 5 days. As shown in Figure 8, GeoDiffusion adapts to diverse geographic constraints and produces reliable short-term forecasts.

**34 days imputation.** This task evaluates imputation performance when nearly five weeks of data are missing after the observed history. In Figure 9, GeoDiffusion successfully completes missing portions while respecting global route structures.

**34 days prediction.** With 29 days of observed data, the next 5 days are predicted. Figure 10 demonstrates that GeoDiffusion effectively extends trajectories under varied regional conditions.

**43 days imputation.** As the missing period grows longer, imputation becomes increasingly challenging. Figure 11 shows that GeoDiffusion maintains consistency with known segments and generates reasonable global-scale continuations.

**43 days prediction.** This setting involves predicting the next 5 days after 38 days of observation. As shown in Figure 12, the generated continuations align with realistic vessel routing, validating GeoDiffusion's robustness under longer horizons.

**60 days imputation.** In the longest imputation scenario, two months of historical data are observed while subsequent long missing intervals are imputed. Figure 13 highlights GeoDiffusion's ability to adaptively reconstruct plausible maritime paths despite large information gaps.

**60 days prediction.** With 55 days of history, GeoDiffusion predicts the next 5 days. As shown in Figure 14, the model captures both stable ocean corridors and flexible coastal patterns, even under extended forecasting horizons.

**Port-to-port planning.** Finally, we evaluate port-to-port trajectory generation. Given origin and destination ports, GeoDiffusion predicts the full connecting trajectory, simulating real-world maritime planning. As illustrated in Figure 4, the predicted blue segments follow realistic global shipping lanes while adapting to different geographic regions and routing demands.

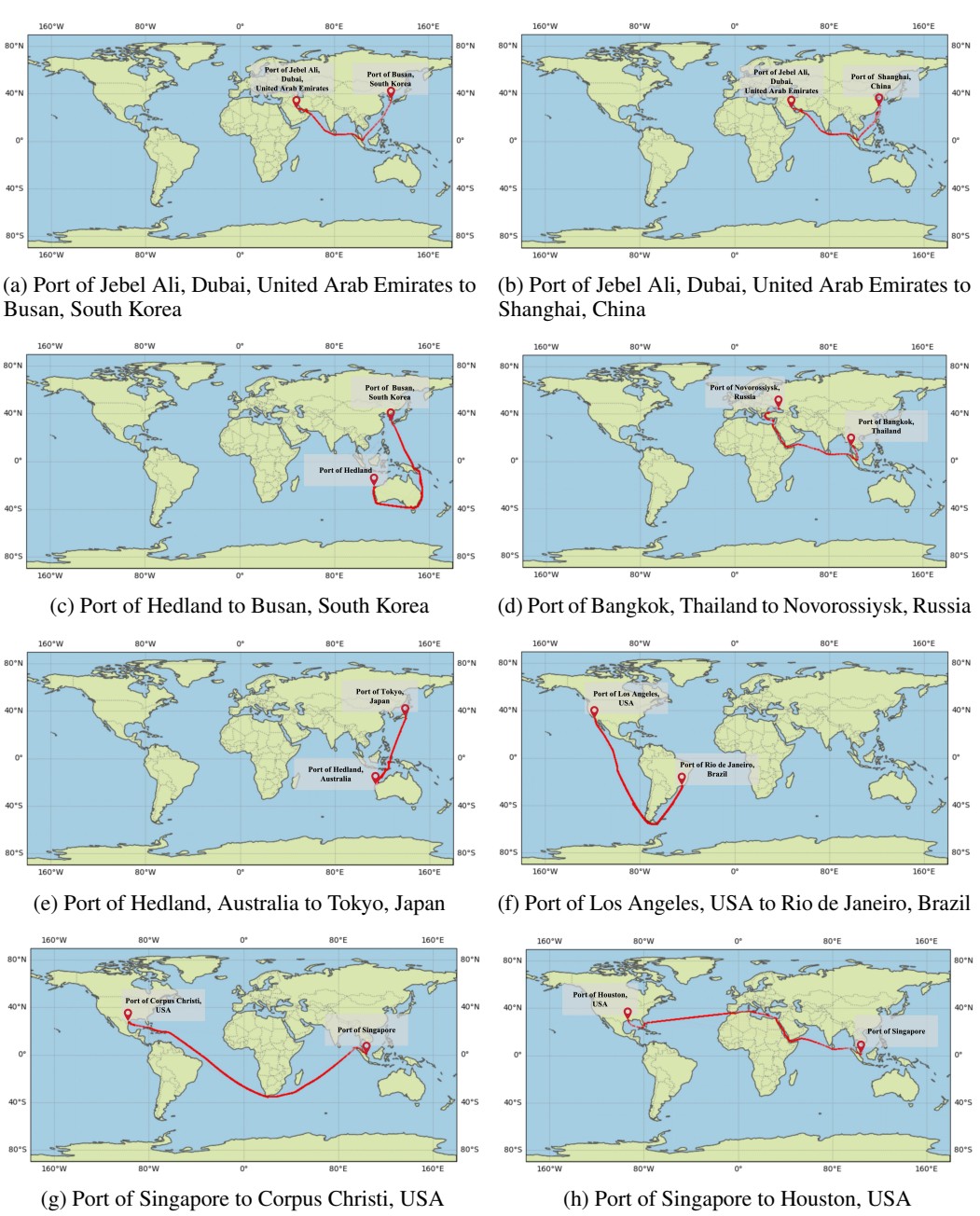

(a) Port of Jebel Ali, Dubai, United Arab Emirates to Busan, South Korea

(b) Port of Jebel Ali, Dubai, United Arab Emirates to Shanghai, China

(c) Port of Hedland to Busan, South Korea

(d) Port of Bangkok, Thailand to Novorossiysk, Russia

(e) Port of Hedland, Australia to Tokyo, Japan

(f) Port of Los Angeles, USA to Rio de Janeiro, Brazil

(g) Port of Singapore to Corpus Christi, USA

(h) Port of Singapore to Houston, USA

Figure 4: Port to Port

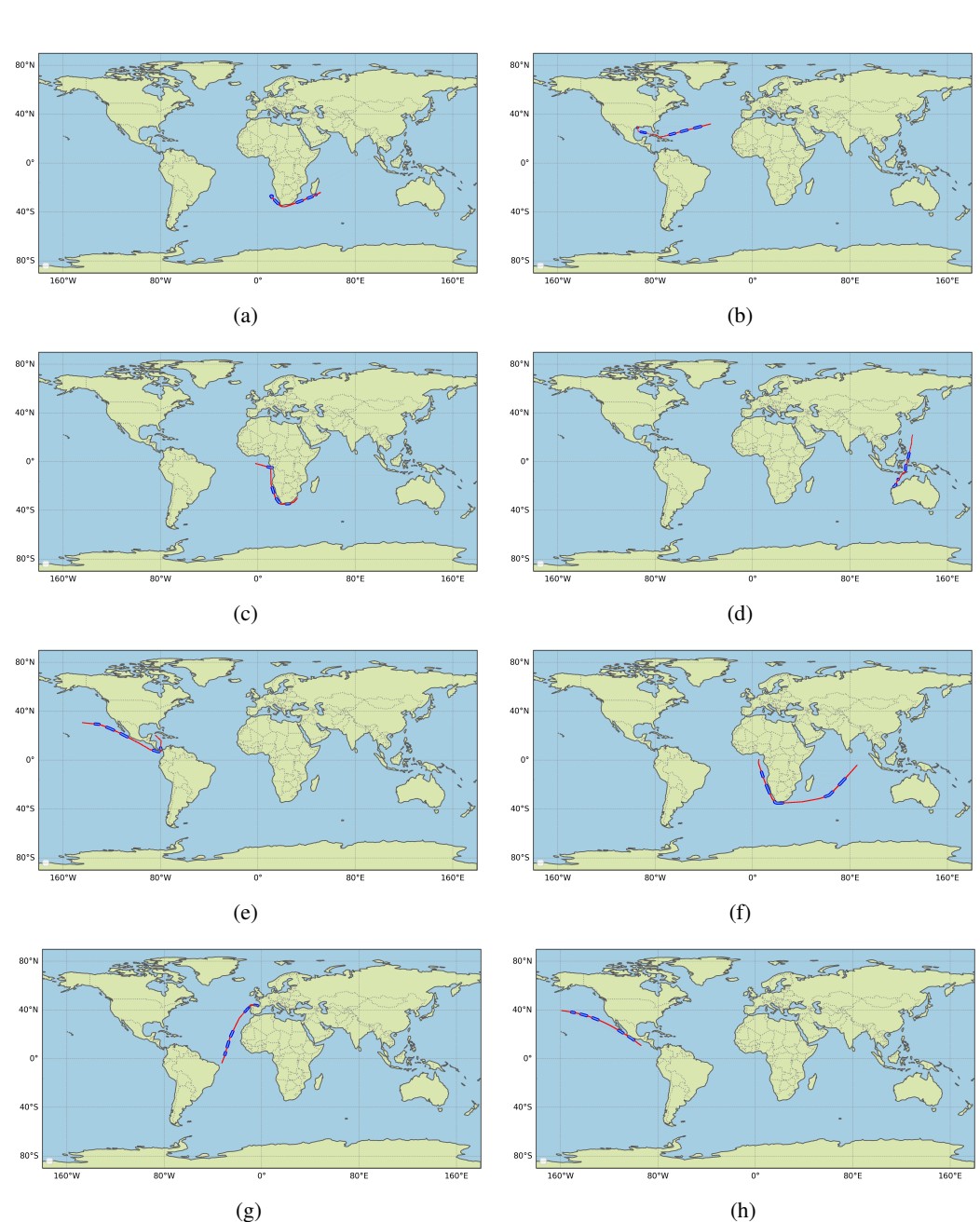

Figure 5: 15days imputation

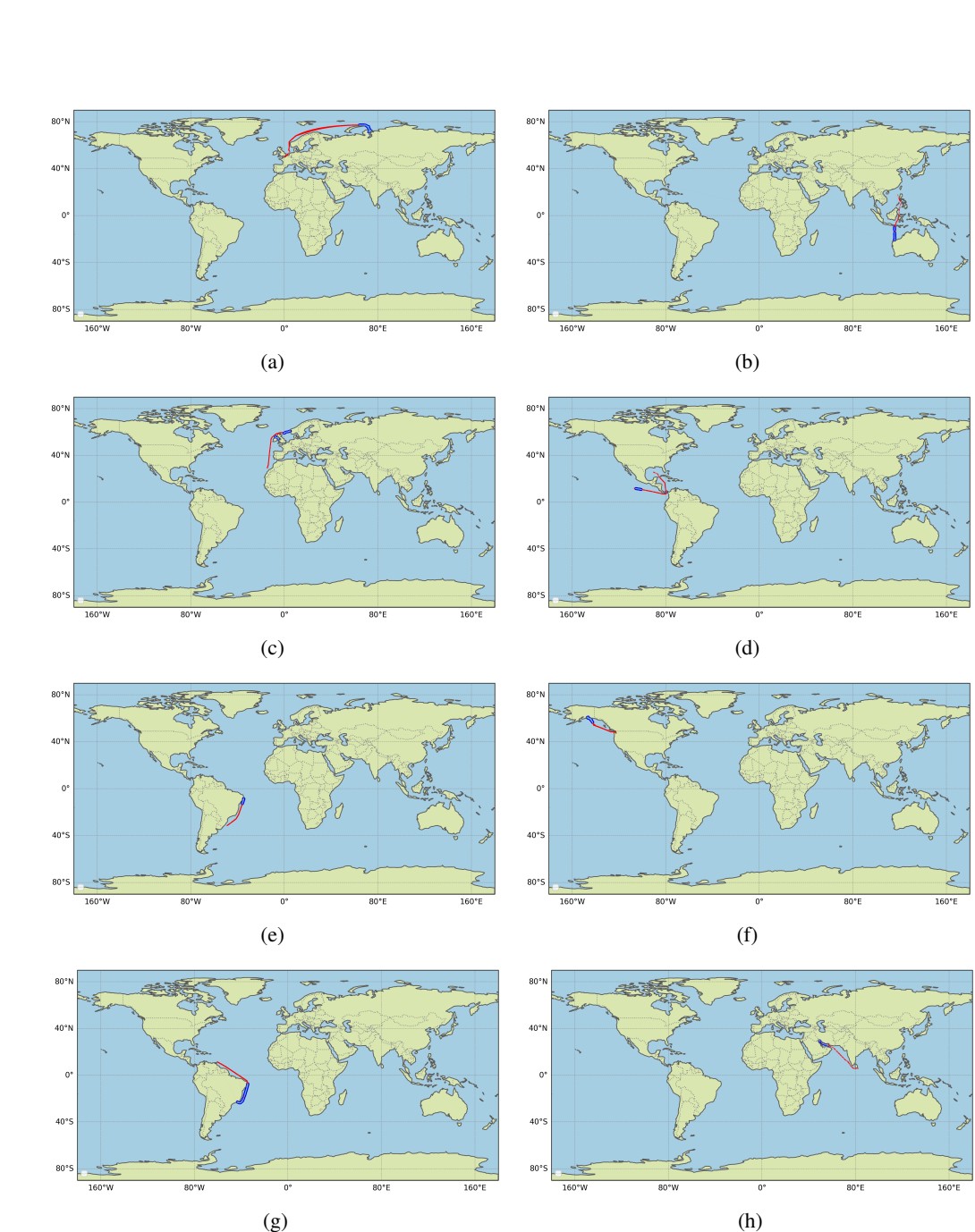

Figure 6: 15 days prediction

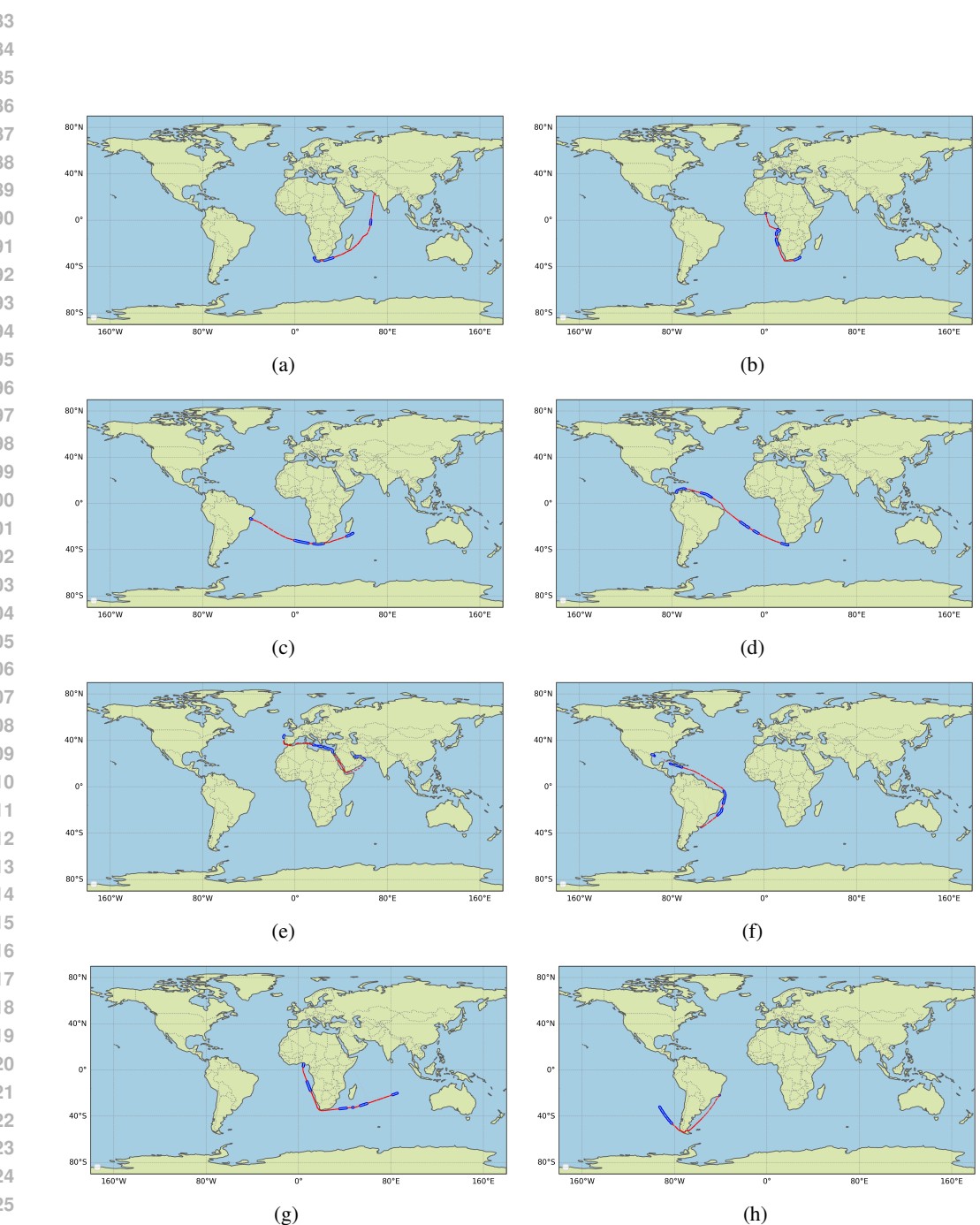

Figure 7: 25 days imputation

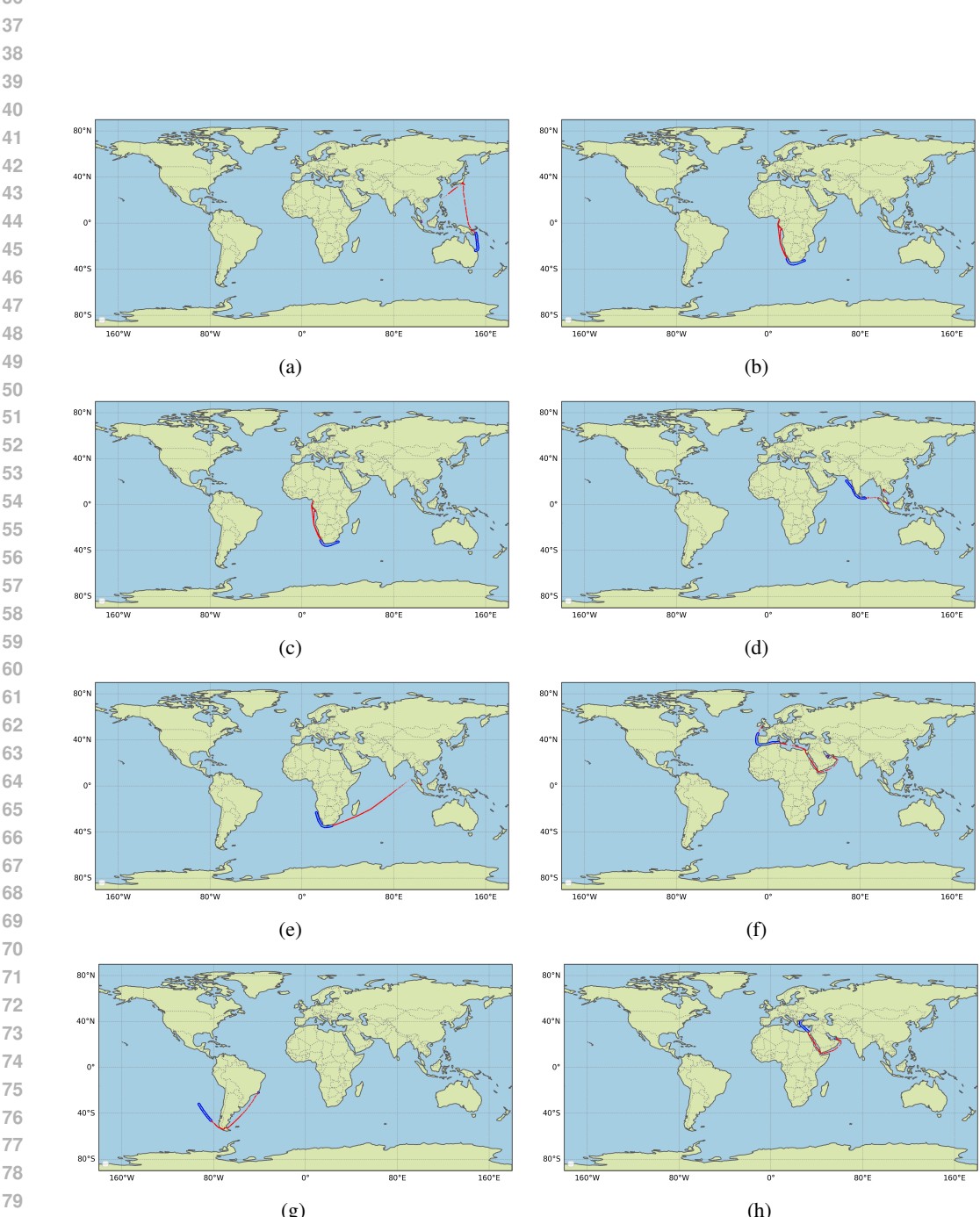

Figure 8: 25 days prediction

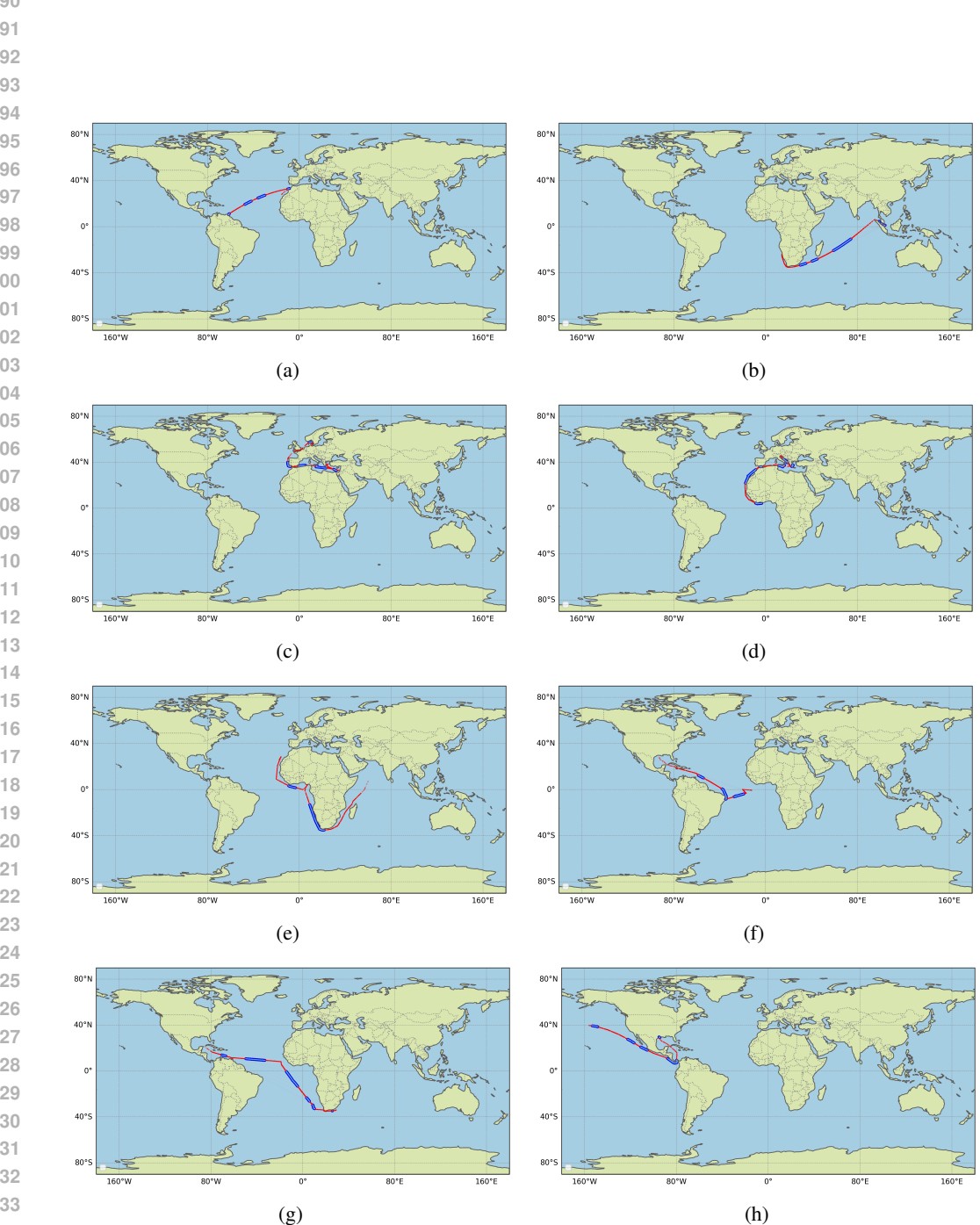

Figure 9: 34 days imputation

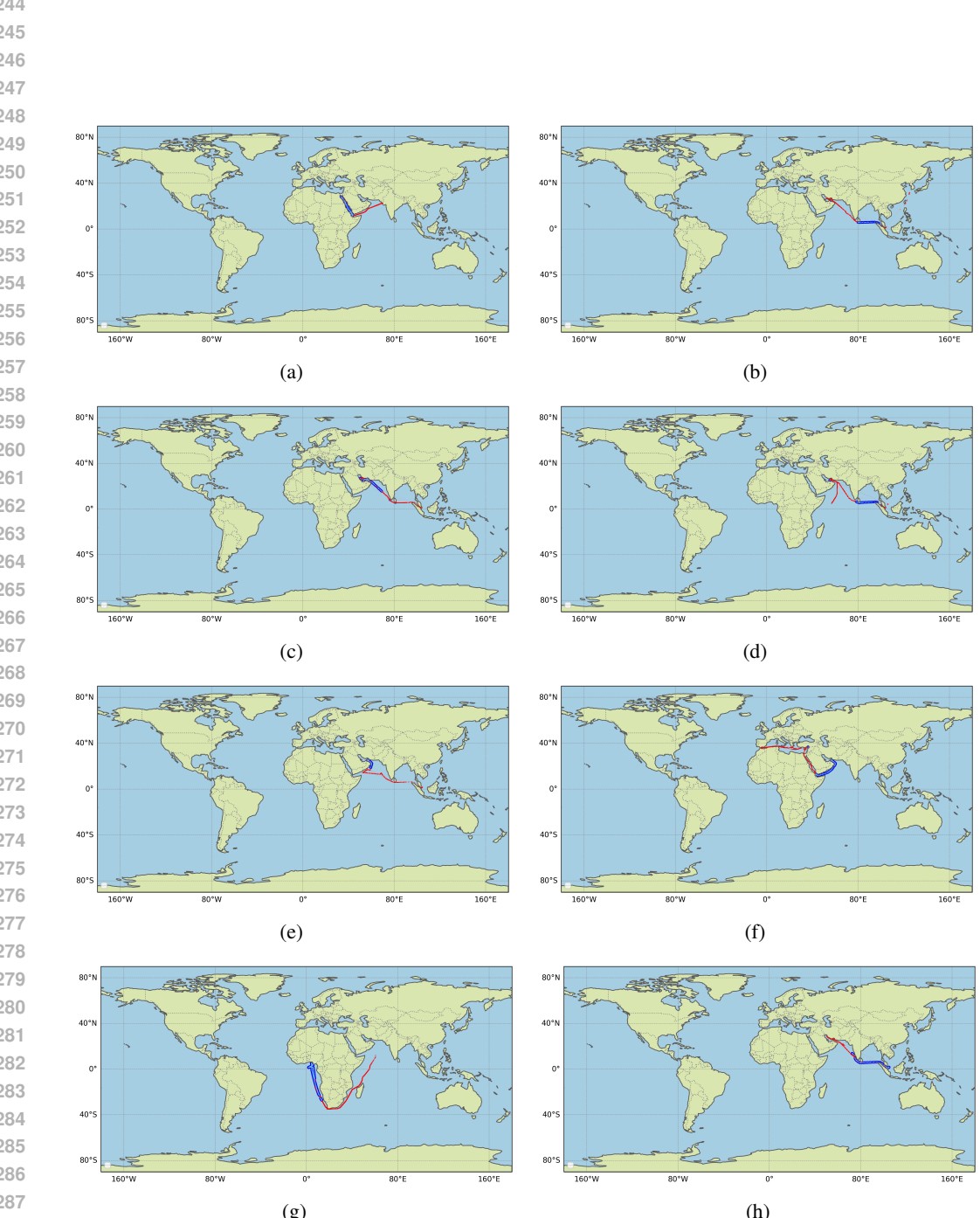

Figure 10: 34 days prediction

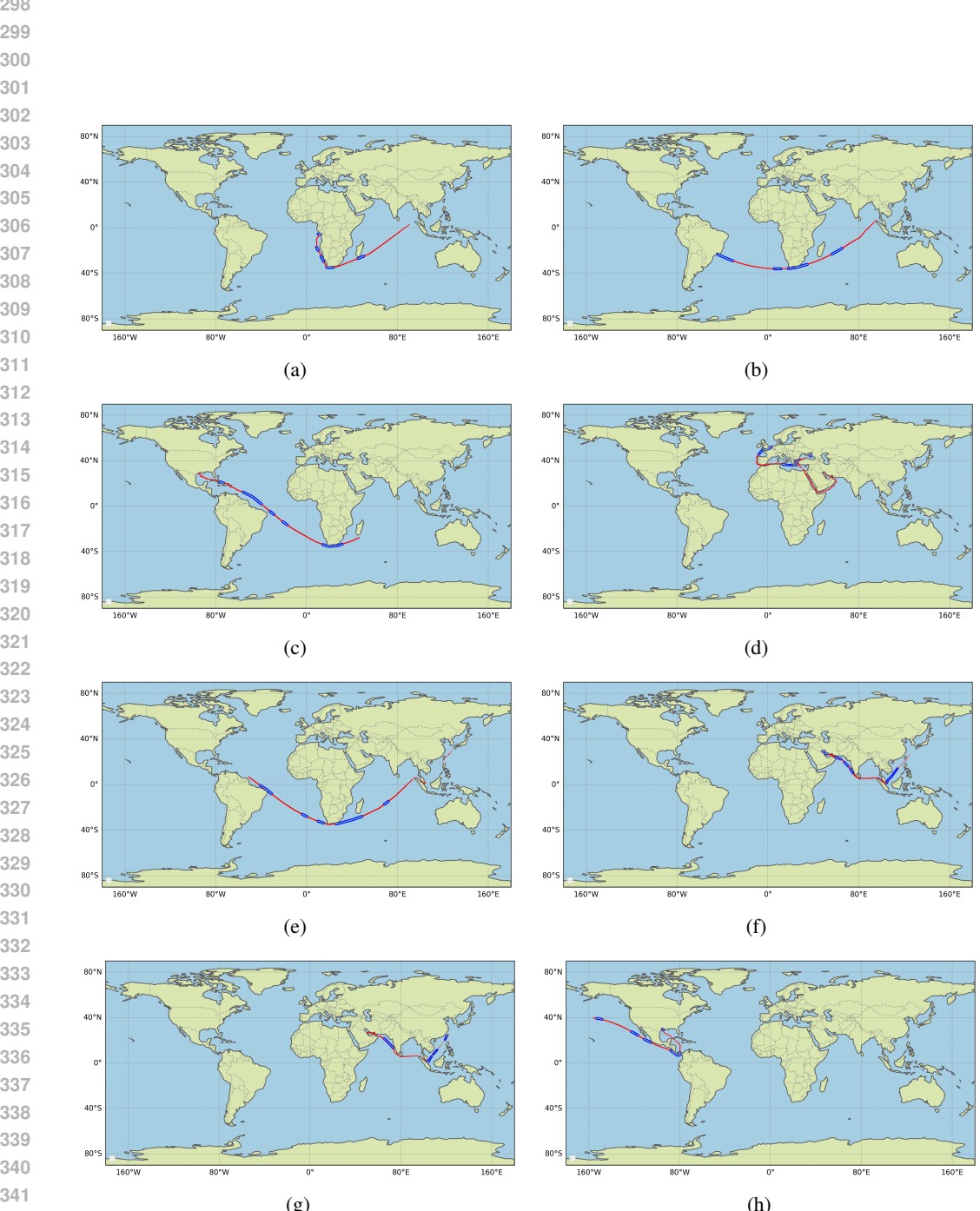

Figure 11: 43 days imputation

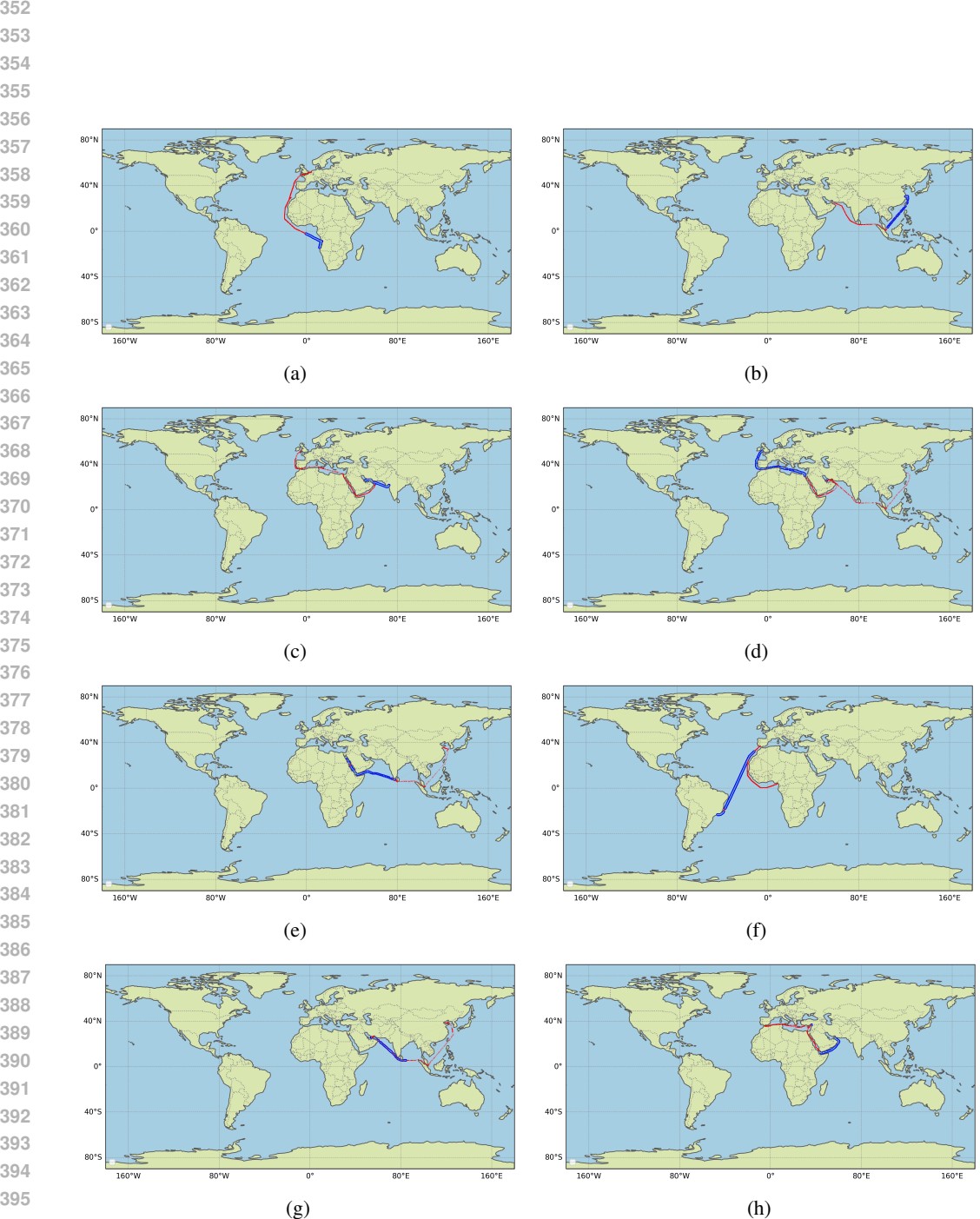

Figure 12: 43 days prediction

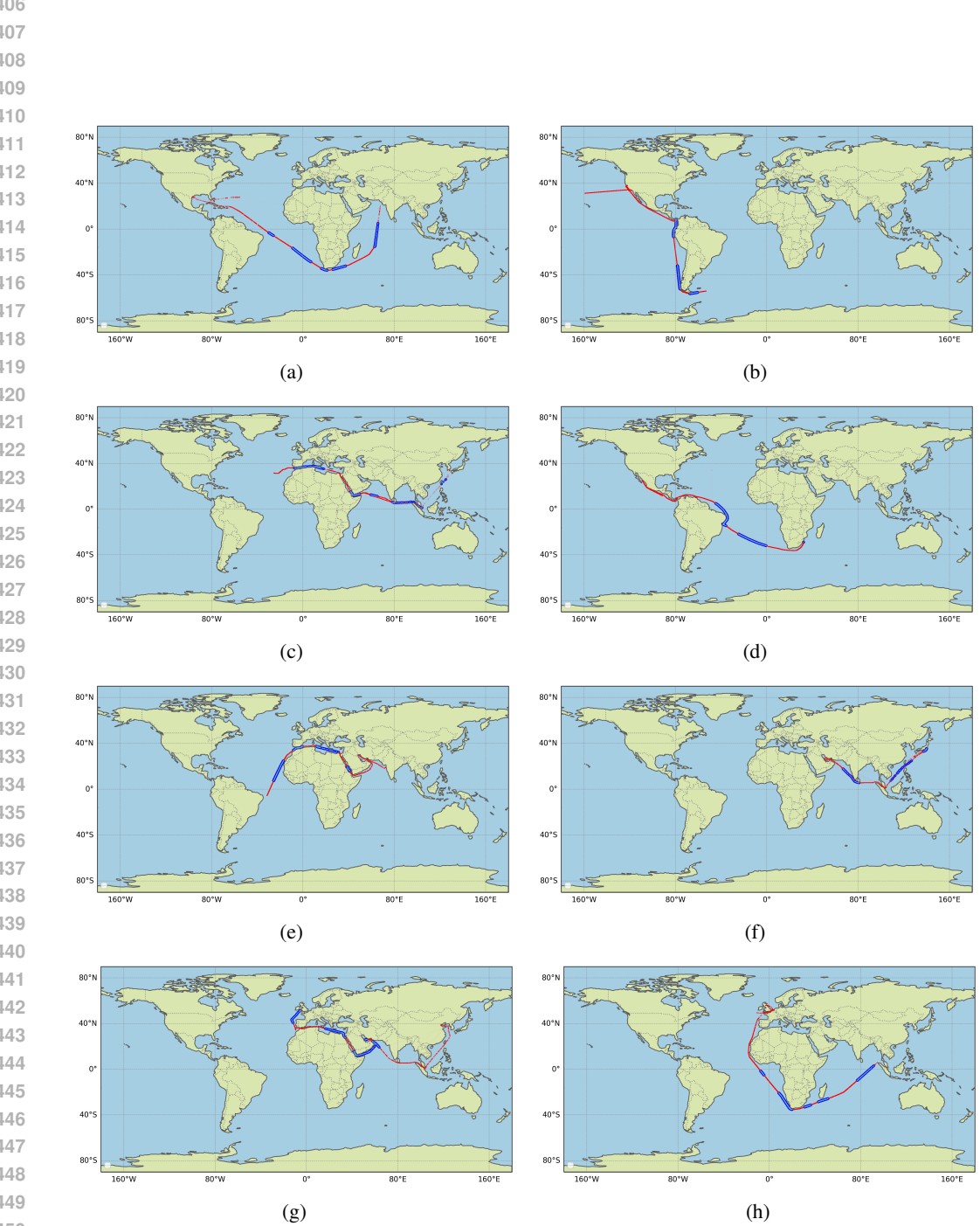

Figure 13: 60 days imputation

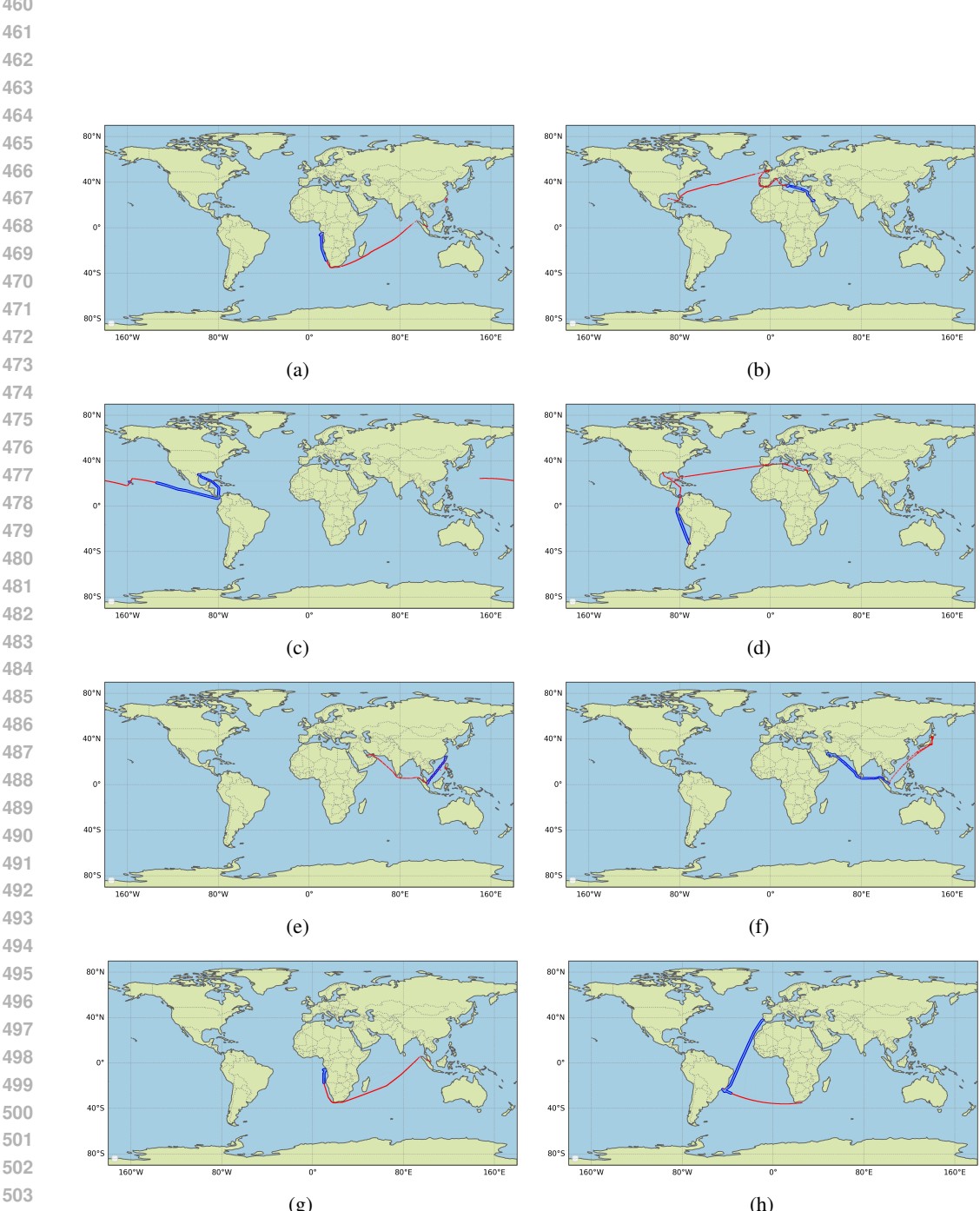

Figure 14: 60 days prediction