# OpenReview forum: "A Diffusion-based Foundation Model for Irregular Spatio-Temporal Trajectories"
_ICLR.cc/2026/Conference — Submitted to ICLR 2026_

### Official Review · Reviewer_RiZC · 2025-10-23

**Soundness:** 3
**Presentation:** 3
**Contribution:** 3
**Rating:** 4
**Confidence:** 4

**Summary:**

This paper proposes a foundation model based on diffusion, named GeoDiffusion, for spatio-temporal sequence modeling in the context of maritime mobility. The model aims to unify three distinct tasks—trajectory prediction, imputation, and route planning—within a single framework. To address the challenges of irregular sampling and missing data inherent in maritime trajectories , the authors introduce a Spatio-Temporal Offset Encoding (STOE) mechanism designed to capture irregular temporal intervals and spatial distances. Furthermore, the model utilizes a Transformer-based denoising network, leveraging self-attention to model the multi-scale dynamics of vessel movements, encompassing both global coarse-grained routes and local fine-grained maneuvers. The unification of diverse tasks is achieved through a training-free conditional inference strategy. The authors conduct extensive experiments across three datasets on all three tasks, reporting that the proposed model achieves state-of-the-art performance under various experimental settings.

**Strengths:**

- The paper is well-written and logically structured. The authors clearly articulate the primary challenges associated with modeling irregular spatio-temporal trajectories, such as data sparsity, multi-scale dynamics, and diverse downstream requirements. The proposed solutions are methodically mapped to these challenges, making the paper easy to follow and understand.

- A key strength of this work lies in its conceptual novelty of framing trajectory prediction, imputation, and route planning as unified generative tasks. By leveraging a diffusion-based generative backbone, the paper successfully proposes a single foundation model that consolidates these diverse objectives , moving beyond the common paradigm of training separate task-specific models. The extensive experimental results confirm that this unified approach is not only novel but also highly effective.

- The introduction of the Spatio-Temporal Offset Encoding (STOE) mechanism is a notable methodological contribution. This component is designed to explicitly address the fundamental problem of irregular sampling and missing observations in spatio-temporal data.

- The paper effectively integrates a conditional diffusion framework to achieve controllable generation, which is crucial for tasks like route planning. By incorporating conditional information, such as departure and destination regions, as control tokens via cross-attention mechanisms, the model can guide the denoising process to produce trajectories that adhere to specified high-level constraints, aligning with the practical needs of maritime applications.

- The development of a training-free inference strategy for trajectory prediction and imputation is a significant practical advantage of this framework. This enhances the model's efficiency and scalability as a true foundation model.

**Weaknesses:**

- The description of the Spatio-Temporal Offset Encoding (STOE) lacks the critical implementation detail for the spatial difference term, $\Delta g_i$ . The authors state its goal is to "capture the true spatial structure" but do not define the calculation method. This ambiguity makes the STOE component non-reproducible. The ablation study shows the module is beneficial, but it is impossible to know if this gain comes from the novel spatial component or just the temporal encoding.

- The paper claims that a Transformer-based network solves the challenge of modeling multi-scale dynamics (global routes vs. local maneuvers) . This claim is unsubstantiated. The justification provided is merely a standard definition of self-attention, not a novel solution or a comparison against other architectures (e.g., U-Net). The evaluations also lack targeted experiments to prove the model's superior performance on this specific multi-scale challenge.

- The paper's validation of geo-temporal and physical constraints is insufficient. A key "Trajectory Constraints" mechanism is cited in the ablation study  as critical for stability, yet this mechanism is never defined in the methodology section. Furthermore, model control is only evaluated via similarity metrics (POR, Jaccard) , not by its adherence to hard constraints (e.g., passing a specific intermediate waypoint), making the robustness of the controlled generation difficult to assess.

- The manuscript contains critical ambiguities that hinder reproducibility. There is a contradiction between the described inference method for imputation (prefix-based, "first L tokens") and the actual experimental setup ("randomly missing days"). It is unclear how the described method handles a random-masking scenario. Additionally, key symbols like $N$ and $l$ have conflicting definitions between the methodology (Section 4.1) and the experimental setup (Section 5.1).

**Questions:**

- To improve clarity, it would be beneficial if the authors could more precisely define the problem setup for each of the three downstream tasks. Could the authors explicitly detail the exact inputs, outputs, and conditioning mechanisms for prediction, imputation, and route planning? For instance, please clarify what constitutes the "condition" and the "clean part" (i.e., the frozen latents) for each distinct task during the diffusion process.

- There appear to be some notational inconsistencies and undefined terms that could be clarified for the reader. For example, the symbol $N$ seems to represent a variable input length in Section 4.1 but a fixed task length in Section 5.1. Could the authors resolve this ambiguity? Similarly, the paper states BART maps variable $N$ tokens to a fixed $l$ , but the mechanism for this is not detailed. A brief explanation would be helpful. Finally, the "Grid-based Trajectory Representation" is central to the method. Could the authors please specify the "spatial discretization function" used and explain how "grid cell IDs" are assigned to "preserve local spatial structure"?

- In the description of the STOE module (Section 4.1), there appears to be a minor notational mismatch. The offset vector $\Delta x$ is defined as $[\Delta s_1; \dots; \Delta s_N]$ (implying $N$ elements), but its dimension is given as $\mathbb{R}^{(N-1)\times2}$ . Could the authors please check and correct this potential inconsistency?

- A key component of the STOE module, the spatial difference term $\Delta g_i$ , is not fully defined. To aid reproducibility, could the authors elaborate on how this term is calculated (e.g., Euclidean distance between grid centers, a learned embedding distance, etc.)? Without this detail, it is difficult to fully assess its contribution. The ablation study (Table 4) effectively shows the STOE module is beneficial, but clarifying the $\Delta g_i$ calculation would help disentangle the specific contributions of the spatial component from the temporal offset component.

- The paper highlights multi-scale modeling as a key challenge and proposes the Transformer's self-attention as the solution. While intuitive, this justification relies on the standard definition of self-attention. Could the authors elaborate on why this standard mechanism is particularly well-suited for maritime multi-scale dynamics, or perhaps discuss if any architectural modifications were made to enhance this capability, beyond just using a standard Transformer?

- Following the previous point, the experimental validation for multi-scale modeling could be strengthened. The current metrics (POR, Jaccard) are generic. Would it be possible for the authors to include a more targeted analysis or experiment (perhaps a qualitative one, or on a specific subset of data) to directly demonstrate the model's superior handling of trajectories that involve both long-range travel and complex near-port maneuvers, as this is a core claim of the paper?

- The concept of conditional control could be specified further. The conditional token $c$ is defined for "departure port or destination region", but its application to prediction and imputation tasks is less clear. Furthermore, the evaluation of control is based on similarity to a ground truth (Table 3). To make the claim of 'control' more robust, could the authors comment on, or perhaps add a simple experiment to assess, the model's ability to adhere to hard constraints (e.g., forcing a trajectory to pass through a specific intermediate region)?

- The rationale for using Classifier-Free Guidance (CFG) was slightly unclear, given that all evaluated downstream tasks are conditional. Could the authors elaborate on the role of the "unconditional task" ( $\mu_{\phi}(z_t, t, \emptyset)$ )? Specifically, how is this unconditional prediction leveraged during sampling for the three conditional tasks evaluated in the paper?

- There seems to be an apparent discrepancy between the methodology and experimental setup for imputation. The methodology (Section 5.1) describes a prefix-based setup ("first L tokens are observed"), which fits prediction, but the imputation experiments (Table 2) mention "randomly missing days". Could the authors please clarify how the described prefix-based inference method is adapted to this non-contiguous, random-masking scenario? Additionally, the ablation study (Table 4) mentions a "trajectory constraint" mechanism. Could the authors define this mechanism in the main text, as it seems crucial for ensuring the physical continuity of generated segments?

---

### Official Review · Reviewer_Awxv · 2025-11-01

**Soundness:** 1
**Presentation:** 2
**Contribution:** 2
**Rating:** 4
**Confidence:** 3

**Summary:**

This paper presents GeoDiffusion, a diffusion-based foundation model for maritime trajectory modeling under irregular and noisy spatio-temporal conditions. It introduces Spatio-Temporal Offset Encoding, latent cross-attention, and a training-free conditional inference mechanism to unify multiple spatio-temporal tasks. Experimental results demonstrate that GeoDiffusion generally performs better than compared baselines.

**Strengths:**

S1. This paper aims to provide a foundation model for irregular spatio-temporal trajectories, which seems interesting and practical.

S2. The paper presents a diffusion-based framework that handles irregular spatio-temporal data through STOE, cross-attention, and a unified training-free inference scheme.

S3. Experiments show the advantages of GeoDiffusion.

**Weaknesses:**

W1: Limited Novelty: The proposed method mainly applies DDPM to trajectory modeling without introducing any specific or novel design tailored to the characteristics of irregular and large-scale trajectories, showing limited methodological originality.

W2: Limited Generalization Across Domains: Despite claims of generality/foundation model, the experiments are restricted to AIS data. The work would be more convincing if evaluated on additional domains mentioned in the introduction, such as urban mobility (e.g., Chengdu or Beijing, which are widely used in existing benchmarks) and air traffic datasets.

W3: Baseline: The baselines for trajectory prediction are mostly from 2021, with only one from 2023. Comparisons with more recent approaches (e.g., [1]) are needed to provide a fair evaluation.
[1] A Diffusion-based Foundation Model for Irregular Spatio-Temporal Trajectories, KDD 2025.

W4: Lack of theoretical Complexity Analysis: The paper does not discuss the algorithmic or computational complexity, making it unclear how the method scales with longer trajectories or larger datasets.

W5: Lack of Computational Efficiency Analysis: The paper omits comparisons of training time, inference latency, and resource consumption against baseline methods, which are essential for assessing real-world applicability.

**Questions:**

Q1: Why were POR and Jaccard Similarity (JS) selected as the evaluation metrics for all three tasks? These metrics mainly measure spatial overlap but may not capture temporal accuracy or uncertainty calibration. Including more standard metrics such as MAE or RMSE could strengthen the evaluation.

Q2: It is suggested to provide the codes and datasets to ensure reproducibility.

Q3: It is better to provide more details regarding the experimental setup, including hardware configuration to improve reproducibility.

Q4: What is the rationale behind selecting the four forecasting horizons (10–5, 23–12, 40–20)? The selection appears somewhat tricky, and it would be helpful to explain how these settings are determined.

---

### Official Review · Reviewer_qrxM · 2025-11-01

**Soundness:** 3
**Presentation:** 2
**Contribution:** 3
**Rating:** 4
**Confidence:** 4

**Summary:**

This paper addresses the significant challenge of modeling maritime AIS trajectories, which are characterized by irregular sampling, noise, and multi-scale dynamics. The authors propose GeoDiffusion, a generative foundation model based on latent diffusion. The central goal is to create a single, pre-trained model that can be applied to three distinct tasks. The model introduces a Spatio-Temporal Offset Encoding (STOE) to manage data irregularity and uses a Transformer backbone to capture long-range and local path dependencies. The authors report state-of-the-art results across these tasks on a private and two public datasets.

**Strengths:**

1. Unified Task Framework: The paper's most significant strength is its "training-free" inference strategy for multi-task unification. The ability to apply a single, pre-trained generative model to diverse tasks like forecasting and planning is a valuable contribution and aligns well with the "foundation model" concept.

2. Qualitative Validation: The visualizations in Figure 3 are a strong supporting element. They demonstrate that the model generates trajectories that are not only statistically accurate but also physically plausible and adhere to real-world geographical constraints.

**Weaknesses:**

1. Numerous Typos and Inconsistencies: The paper suffers from a lack of careful proofreading, which creates confusion and hinders understanding.
    - Equation (3) defines $\Delta x$ as a sequence of $N$ tokens, $[\Delta s_{1};...;\Delta s_{N}]$, but gives its dimension as $(N-1)\times2$. This is a contradiction.
    - In the description of Equation (3) (line 225), the text refers to the temporal gap as the "third term," but the equation itself only lists two terms.
     - Equation (4) introduces a variable $\Delta v_{ij}$ that is never defined in the text.
2. There are quite a few issues with tables where bolding and underlining are not consistent for all methods, including Tab. 1, Tab. 2, and Tab. 3.
3. Missing Key Citations: The related work section is incomplete and misses several recent and highly relevant publications in trajectory modeling and imputation, such as [1] and [2].

[1] "Micro-macro spatial-temporal graph-based encoder-decoder for map-constrained trajectory recovery" (TKDE 2024).

[2] "ProDiff: Prototype-Guided Diffusion for Minimal Information Trajectory Imputation" (ICML 2025).

**Questions:**

Q1: Motivation for Diffusion Model. While Section 4.2 clearly motivates the use of a Transformer, the methodology section itself provides no clear motivation for choosing a diffusion model as the generative backbone. The justification in the introduction should be more clearly connected to the methodological description.

Q2: Choice of Metrics. The evaluation relies solely on two grid-based metrics: POR and Jaccard Similarity . Are these metrics comprehensive enough?  The authors should add more metrics or provide a strong justification for why POR and Jaccard are sufficient.

Q3: Sensitivity to Grid Resolution. The entire method is built on discretizing continuous coordinates into a grid. The chosen grid resolution (which is never stated) would seem to have a significant impact on the task's difficulty and the resulting metrics. Is the model's performance robust to this choice? The paper would be much stronger with an ablation study analyzing the effect of different grid resolutions.

---

### Official Review · Reviewer_A6U1 · 2025-11-01

**Soundness:** 2
**Presentation:** 3
**Contribution:** 2
**Rating:** 4
**Confidence:** 4

**Summary:**

This paper introduces GeoDiffusion, a diffusion-based foundation model for modeling irregular spatio-temporal trajectories. It uses global maritime AIS (Automatic Identification System) data as a challenging testbed, which is inherently noisy, irregularly sampled, and involves multi-scale dynamics and geographical constraints. The core contribution is a unified generative framework that can handle multiple downstream tasks. To achieve this, the model introduces three key components: Spatio-Temporal Offset Encoding (STOE), Transformer Denoising Network and Training-Free Conditional Inference.

**Strengths:**

1.	The Spatio-Temporal Offset Encoding is a clever innovation for irregular time-series data. Standard positional encodings struggle with variable time gaps and spatial jumps, but STOE explicitly models these offsets, providing the model with a crucial inductive bias.
2.	The model is validated on three distinct and practical tasks.
3.	GeoDiffusion achieves top performance consistently across all datasets and tasks, as shown in Tables 1, 2, and 3.

**Weaknesses:**

1.	Overstated "Foundation Model" Claim: The paper's central claim of building a "foundation model" is somewhat overstated. The model is pre-trained only on tanker trajectories, which represents a narrow domain. While evaluated on two public datasets, these are still within the maritime domain. The paper claims its solutions generalize to other domains (e.g., urban mobility), but provides no experimental evidence for this. A more accurate description might be a "foundation model for maritime tanker trajectories."
2.	Key Methodological Details are Missing: The definition of the core STOE component is vague. The paper defines the offset vector as $\Delta s_i = [\Delta g_i, log(1+|\tau_{i+1}-\tau_i|)]$. However, it never provides a clear mathematical definition for $\Delta g_i$ (the spatial offset). It only states that $\Delta g_i$ accounts for differences between grid IDs.
3.	Reproducibility: The model relies on a large, private dataset for pre-training, which limits the ability of the research community to reproduce the pre-training stage.

**Questions:**

•  Q1: Unclear Data Description: The paper lacks a clear description of the AIS data, especially the pre-training dataset. What is the scale (e.g., number of trajectories, total points)? How was the data pre-processed?
• Q2: Confusing Ablation Study: The ablation study in Table 4 includes a setting "w/o Trajectory Constraints". This term, "Trajectory Constraints," is never defined in the methodology section (Section 4). What does this component refer to? Is it the prefix conditioning (freezing known tokens) used during inference? Is it some other physical constraint?

---

### Meta-Review · Area_Chair_CCxj · 2026-01-08

**Summary:**

- Overstated “foundation model”/generality claims (A6U1, Awxv): Pretraining and evaluation are largely confined to maritime AIS, with no evidence of cross-domain transfer despite claims (e.g., urban mobility).

- Reproducibility and technical ambiguity (A6U1, RiZC, qrxM): Key components are underspecified or inconsistent, especially STOE’s spatial offset term, undefined symbols, and contradictions between methodology and experiments. Reliance on a private pretraining dataset further limits reproducibility.

- Evaluation rigor and fairness gaps (Awxv, qrxM, RiZC): Concerns about outdated/missing baselines, incomplete related work, and insufficient validation of claimed properties (multi-scale dynamics, constraint adherence). Missing complexity/efficiency analysis (training/inference cost, scalability).

The above issues support a decision to reject.

**Reviewer Concerns:**

No response provided.

**Reviewer Scores:**

No response provided.

---

### Decision · Program_Chairs · 2026-01-26

Reject